# PERCEIVER IO: A GENERAL ARCHITECTURE FOR STRUCTURED INPUTS & OUTPUTS

**Andrew Jaegle, Sebastian Borgeaud, Jean-Baptiste Alayrac, Carl Doersch, Catalin Ionescu,**

**David Ding, Skanda Koppula, Daniel Zoran, Andrew Brock, Evan Shelhamer, Olivier Hénaff,**

**Matthew M. Botvinick, Andrew Zisserman, Oriol Vinyals, João Carreira**

DeepMind

## ABSTRACT

A central goal of machine learning is the development of systems that can solve many problems in as many data domains as possible. Current architectures, however, cannot be applied beyond a small set of stereotyped settings, as they bake in domain & task assumptions or scale poorly to large inputs or outputs. In this work, we propose Perceiver IO, a general-purpose architecture that handles data from arbitrary settings while scaling linearly with the size of inputs and outputs. Our model augments the Perceiver with a flexible querying mechanism that enables outputs of various sizes and semantics, doing away with the need for task-specific architecture engineering. The same architecture achieves strong results on tasks spanning natural language and visual understanding, multi-task and multi-modal reasoning, and StarCraft II. As highlights, Perceiver IO outperforms a Transformer-based BERT baseline on the GLUE language benchmark despite removing input tokenization and achieves state-of-the-art performance on Sintel optical flow estimation with no explicit mechanisms for multiscale correspondence.

## 1 INTRODUCTION

Humans have a remarkable ability to take in data from many sources, integrate it seamlessly, and deploy it in the service of a range of goals. Most machine learning research focuses on building bespoke systems to handle the stereotyped inputs and outputs associated with a single task. This is true even for models that handle multiple modalities. A typical approach independently processes each input with a modality specific architecture (for example using a 2D ResNet (He et al., 2016) for vision and a Transformer (Vaswani et al., 2017) for language), integrates them afterwards using a third fusion network, and reads out the result in a task-specific manner. The complexity of systems like this can grow dramatically as the inputs or outputs grow more diverse (e.g. Abramson et al. 2020; Vinyals et al. 2019; Ramesh et al. 2021), and the structure of a task's inputs and outputs may place strong constraints on how data is processed, making adaptation to new settings difficult.

Is the development of problem-specific models for each new set of inputs and outputs unavoidable? Life would be drastically simpler if a single neural network architecture could handle a wide variety of both input modalities and output tasks. In this work, we propose such an architecture, with the ultimate goal of building a network that can easily integrate and transform arbitrary information for arbitrary tasks. Our starting point is the Perceiver (Jaegle et al., 2021), an architecture which has demonstrated a remarkable ability to handle data from many modalities with no changes to the network architecture. The Perceiver uses attention to map inputs of a wide range of modalities to a fixed-size latent space that is further processed by a deep, fully attentional network. This process decouples the bulk of the network's processing from the size and modality-specific details of the input, allowing it to scale to large and multimodal data.

But the Perceiver can only handle simple output spaces like classification. Much of the complexity of real-world tasks comes from the variety, size, and structure of their *outputs*, and in this regard

Figure 1: The Perceiver IO architecture can be used on domains with a wide variety of input and output spaces, including multi-task language understanding, dense visual tasks like optical flow, hybrid dense/sparse multimodal tasks such as video+audio+class autoencoding, and tasks with symbolic outputs like StarCraft II. See Tables 5 and 6 for details of all domains considered here.

the original Perceiver can't be considered general purpose. In this work, we develop a mechanism for decoding structured outputs – language, optical flow fields, audiovisual sequences, symbolic unordered sets, etc. – directly from the Perceiver latent space, which allows the model to handle a host of new domains without sacrificing the benefits of deep, domain-agnostic processing. To do this, we produce each output by attending to the latent array using an *output query* that specifies the semantics of that particular output. For example if we wanted the model to predict optical flow on one particular pixel we could compose a query from the pixel's xy coordinates plus an optical flow task embedding: the model would then attend using the query and produce a single flow vector. As a result, our architecture can produce many outputs, each with arbitrary shape and structure, and yet the latent features in our architecture remain agnostic to the shape and structure of the outputs.

Perceiver IO does this using a fully attentional read-process-write architecture: inputs are encoded (read) to a latent space, the latent representation is refined (process) via many layers of processing, and the latent space is decoded (write) to produce outputs. This approach inherits the best features of both Transformers – which leverage domain agnostic primitives for nonlocal processing of inputs – and the encoder-decoder architectures (e.g. Ronneberger et al. 2015; Newell et al. 2016) that are in widespread use in high-bandwidth domains such as computer vision or multimodal processing. This approach allows us to decouple the size of elements used for the bulk of the computation (the latent) from the size of the input and output spaces, while making minimal assumptions about the spatial or locality structure of the input and output.

Perceiver IO's decoding procedure uses an attention mechanism to map from latents to arbitrarily sized and structured outputs using a querying system that can flexibly specify the semantics needed for outputs on a wide range of domains, including dense and multitask settings. This decoder allows Perceiver IO to serve as a drop-in replacement for a wide range of specialist networks currently in use on a set of challenging domains, while improving performance on tasks like classification that could be handled by the Perceiver.

The proposed architecture can be applied with unprecedented levels of generality. Perceiver IO can replace the Transformers used in BERT (Devlin et al., 2019) and AlphaStar (Vinyals et al., 2019). At the same time, Perceiver IO produces state-of-the-art results on the Sintel optical flow benchmark (Butler et al., 2012) and good results on ImageNet image classification (Deng et al., 2009). Perceiver IO produces compelling results even when handling highly diverse multimodal data, such as on joint {video, audio, label} autoencoding in Kinetics (Smaira et al., 2020) and joint audio-video classification on AudioSet (Gemmeke et al., 2017). Perceiver IO allows us to simplify pipelines and remove domain-specific assumptions: we process language without tokenizers without a performance or speed hit, fine-tune on multiple classification tasks simultaneously and without the need for `[CLS]` tokens (Sec. 4.1), estimate optical flow without relying on explicit architectural features for multiscale correspondence (Sec. 4.2), learn joint representations of video, audio, and labels without separate network trunks (Sec. 4.3), and perform image classification with no information about the 2D structure of images (Sec. A).

## 2 RELATED WORK

Neural network research has long sought architectures that can handle large, arbitrarily structured inputs and outputs. Autoencoding (Hinton & Zemel, 1994) was among the first attempts to build

representation which could encode and reproduce high-dimensional inputs like images. As hardware grew more powerful, neural nets led to breakthroughs in image understanding (Krizhevsky et al., 2012; Zeiler & Fergus, 2014; Szegedy et al., 2015) and interest intensified: autoregressive models that could process and complete samples of handwriting were developed (Graves, 2013), and new convolutional network designs led to good results in structured output spaces like semantic segmentation (Farabet et al., 2012; Long et al., 2015; Ronneberger et al., 2015), pose estimation (Toshev & Szegedy, 2014), detection (Sermanet et al., 2014), captioning (You et al., 2016), and optical flow (Fischer et al., 2015). At the same time, natural language applications research has made extensive progressive in capturing the structured nature of language, typically via autoregressive models (Collobert et al., 2011; Sutskever et al., 2014; Vaswani et al., 2017; Radford et al., 2019; Brown et al., 2020) or context prediction (Mikolov et al., 2013; Pennington et al., 2014; Devlin et al., 2019).

Similar to our work, several groups have proposed to solve tasks in multiple domains (e.g. Kaiser et al. 2017; Alayrac et al. 2020; Akbari et al. 2021), but typically across a fixed and predefined set of modalities by means of domain-specific networks. Although single-task specialist networks remain dominant in vision, multi-task learning has become popular (Misra et al., 2016; Doersch & Zisserman, 2017; Kokkinos, 2017; Zamir et al., 2018) and individual models achieve generality in a restricted domain: e.g. Mask-RCNN (He et al., 2017) handles object detection, segmentation, and pose estimation. In language, training or evaluation on multiple tasks has also become common (Collobert & Weston, 2008; Luong et al., 2016; Devlin et al., 2019; Liu et al., 2019; Raffel et al., 2020). Several groups have demonstrated that Transformers (originally designed for language) can be used or adapted to non-language tasks (e.g. Chen et al. 2020; Lu et al. 2021), but the limited scalability of Transformers limits their usefulness as general-purpose architectures.

Several groups have proposed to use attention to manipulate the size of arrays or to introduce bottlenecks in processing. Set Transformers and related work (Lee et al., 2019; Goyal et al., 2022) use a learned query ("inducing points") to induce local bottlenecks by mapping a set back and forth from a set with fewer elements and learned decoder queries ("seed vectors") to map to outputs ("pooling by multiheaded attention"). Each layer of these networks has complexity linear in the input size, while Perceivers use a deep latent network with complexity independent of the input and output. Our work uses attention over inputs and outputs of different sizes in part to produce an efficient attention architecture, and several other efficient attention architectures have been proposed, largely for language or small-scale problems (e.g. Xiong et al. 2021; Wang et al. 2020; Tay et al. 2021a; Beltagy et al. 2020 and see Tay et al. 2021b). The focus of our work is developing an architecture that is efficient and also performs well in many settings with a wide range of inputs and outputs. Several works use attention to process latent spaces that interface with input/output data using task- or domain-specific architectures (Carion et al., 2020; Locatello et al., 2020; Wang et al., 2021), and cross-attention itself is widely used to produce outputs in of a different size or structure from inputs (Dai et al., 2019; Desai & Johnson, 2021; Miech et al., 2021; Vaswani et al., 2017; Raffel et al., 2020; Santoro et al., 2018; Hudson & Zitnick, 2021; Ma et al., 2021). Perceiver IO builds on this body of work to produce a general purpose architecture that can be easily and widely applied.

## 3    THE PERCEIVER IO ARCHITECTURE

The Perceiver IO architecture builds on the Perceiver (Jaegle et al., 2021), which achieved its cross-domain generality by assuming that its input is a simple 2D byte array: a set of elements (which might be pixels or patches in vision, characters or words in language, or some form of embedding, learned or otherwise), each described by a feature vector. The model then encodes information about the input array using a smaller number of latent feature vectors, using Transformer-style attention, followed by iterative processing and a final aggregation down to a category label.

Rather than output a single category, Perceiver IO aims to have the same level of generality with respect to its *outputs* as the Perceiver has with respect to its *inputs*: that is, it should produce arbitrary output arrays. We can predict each element of the output array using another attention module by *querying* the latent array using a query feature vector unique to the desired output element. In other words, we define a query array with the same number of elements as the desired output. The queries may be hand-designed, learned embeddings, or a simple function of the input. They attend to the latents to yield an output array of the desired shape.

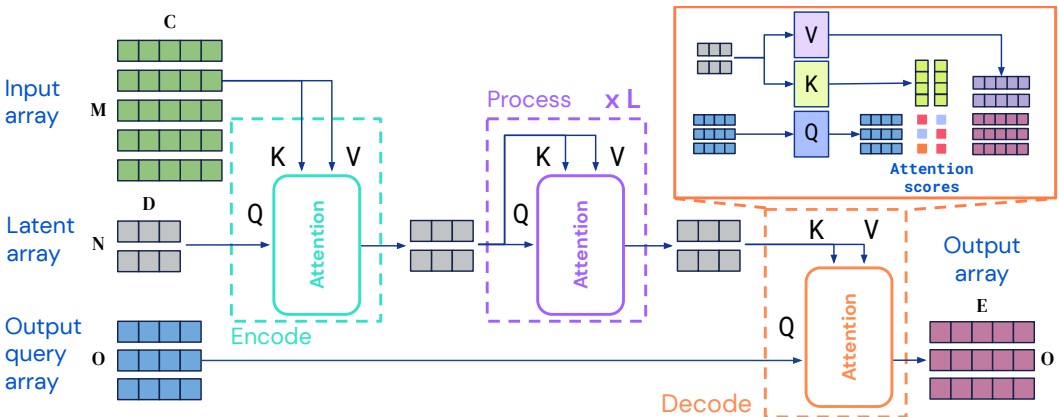

Figure 2: The Perceiver IO architecture. Perceiver IO maps arbitrary input arrays to arbitrary output arrays in a domain agnostic process. The bulk of the computation happens in a latent space whose size is typically smaller than the inputs and outputs, which makes the process computationally tractable even for very large inputs & outputs. See Fig. 5 for a more detailed look at encode, process, and decode attention.

### 3.1 ENCODING, PROCESSING, AND DECODING

Fig. 2 illustrates the Perceiver IO. We first **encode** by applying an attention module that maps input arrays $x \in \mathbb{R}^{M \times C}$ to arrays in a latent space $z \in \mathbb{R}^{N \times D}$. We next **process** the latents $z$ by applying a series of modules that take in and return arrays in this latent space. Finally, we **decode** by applying an attention module that maps latent arrays to output arrays $y \in \mathbb{R}^{O \times E}$. $M$, $C$, $O$, and $E$ are properties of the task data and can be very large (Tab. 5), while $N$ and $D$ are hyperparameters and can be chosen to make model computation tractable. Following the design of the Perceiver, we implement each of the architecture's components using Transformer-style attention modules.

Each of these modules applies a global query-key-value (QKV) attention operation followed by a multi-layer perceptron (MLP). As usual in Transformer-style architectures, we apply the MLP independently to each element of the index dimension. Both encoder and decoder take in two input arrays, the first used as input to the module's key and value networks, and the second used as input to the module's query network. The module's output has the same index dimension (the same number of elements) as the query input.

The Perceiver IO architecture builds on primitives similar to those in Transformers. Why aren't Transformers all you need? Transformers scale very poorly in both compute and memory (Tay et al., 2020). Because Transformers deploy attention modules homogeneously throughout its architecture, using its full input to generate queries and keys at every layer. This means each layer scales quadratically in compute and memory, which makes it impossible to apply Transformers on high-dimensional data like images without some form of preprocessing. Even on domains like language where Transformers shine, preprocessing (e.g. tokenization) is often needed to scale beyond short input sequences. Perceiver IO uses attention non-homogeneously by mapping inputs to a latent space, processing in that latent space, and decoding to an output space. Perceiver IO has no quadratic dependence on the input or output size: encoder and decoder attention modules depend linearly on the input and output size (respectively), while latent attention is independent of both input and output sizes (Sec. E.2). Because of the corresponding reduction in compute and memory requirements, Perceiver IO scales to much larger inputs and outputs. While Transformers are typically used in settings with data preprocessed to contain at most a few thousand dimensions (Brown et al., 2020; Raffel et al., 2020), we show good results on domains with hundreds of thousands of dimensions.

This architecture can be applied to inputs of any shape or spatial layout including inputs or outputs with different spatial structure (e.g. sound and video). In contrast to latent spaces typically used in vision (e.g. Ronneberger et al. 2015) the latent does not explicitly share the structure (spatial or otherwise) of the inputs. To decode this information, we query for it using cross-attention.

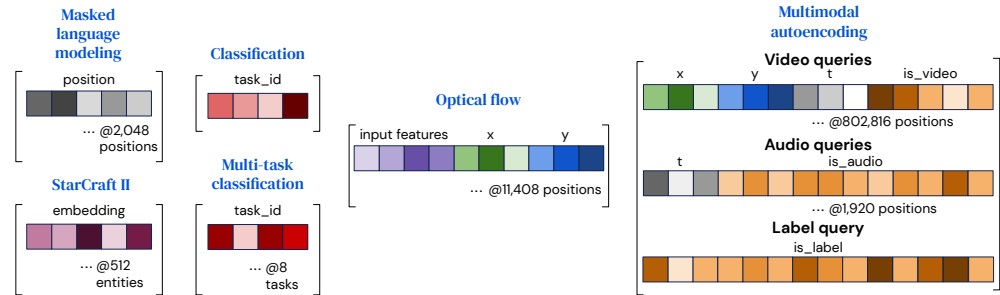

Figure 3: We construct queries with output-specific features to produce outputs with different semantics. For settings where each output point differs only in its position, like language, a position embedding can be used. Input features for the target output can also be used to query, either alone (as for StarCraft II) or alongside position features (as for flow). For multi-{task, modal} settings we use one embedding for each {task, modality} instead of each position. A single learned embedding suffices for simple classification tasks, like ImageNet. For tasks with heterogeneous outputs like multimodal autoencoding, features that are specific to some queries (like xy position) can be combined with modality embeddings, which also pad embeddings to fixed length.

## 3.2 DECODING THE LATENT REPRESENTATION WITH A QUERY ARRAY

Our goal is to produce a final output array of size $O \times E$, given a latent representation of size $N \times D$. We produce an output of this size by querying the decoder with an array of index dimension $O$. To capture the structure of the output space, we use queries containing the appropriate information for each output point, e.g. its spatial position or its modality.

We construct queries by combining (concatenating or adding) a set of vectors into a query vector containing all of the information relevant for one of the $O$ desired outputs. This process is analogous to the way that positional information is used to query implicit functions like NeRF (Mildenhall et al., 2020). We illustrate the query structure for the tasks we consider here in Fig. 3. For tasks with simple outputs, such as classification, these queries can be reused for every example and can be learned from scratch. For outputs with a spatial or sequence structure, we include a position encoding (e.g. a learned positional encoding or a Fourier feature) representing the position to be decoded in the output. For outputs with a multi-task or multimodal structure, we learn a single query for each task or for each modality: this information allows the network to distinguish one task or modality query from the others, much as positional encodings allow attention to distnguish one position from another. For other tasks, the output should reflect the content of the input at the query location: for instance, for flow we find it helpful to include the input feature at the point being queried, and for StarCraft II we use the unit information to associate the model's output with the corresponding unit. We find that even very simple query features can produce good results, suggesting that the latent attention process is able to learn to organize the relevant information in a way that's easy to query.

Each output point depends only on its query and the latent array, allowing us to decode outputs in parallel. This property allows us to amortize model training on datasets of very large output size. For example, Kinetics consists of labels, video voxels, and audio samples which together come to over 800,000 points (Tab. 5), which is prohibitively expensive to decode at once, even with linear scaling. Instead, we subsample the output array at training time and compute the loss on an affordable subset of points. At test time, we generate outputs in batches to produce the full output array.

## 4 EXPERIMENTS

To probe the generality of Perceiver IO, we evaluate it on several domains including language understanding (Wikipedia+C4 masked language modeling), visual understanding (Sintel/KITTI optical flow and ImageNet classification), multi-modal (Kinetics autoencoding and AudioSet classification) & multi-task settings (multi-task GLUE), and symbolic representations for games (StarCraft II). All experiments were conducted using JAX (Bradbury et al., 2018) and the DeepMind JAX ecosystem (Babuschkin et al., 2020).

| Model | Tokenization | $M$ | $N$ | Depth | Params | FLOPs | SPS | Avg. |
|---|---|---|---|---|---|---|---|---|
| BERT Base (test) | SentencePiece | 512 | 512 | 12 | 110M | 109B | - | 81.0 |
| BERT Base (ours) | SentencePiece | 512 | 512 | 12 | 110M | 109B | 7.3 | 81.1 |
| Perceiver IO Base | SentencePiece | 512 | 256 | 26 | 223M | 119B | 7.4 | **81.2** |
| BERT (matching FLOPs) | UTF-8 bytes | 2048 | 2048 | 6 | 20M | 130B | 2.9 | 71.5 |
| Perceiver IO | UTF-8 bytes | 2048 | 256 | 26 | 201M | 113B | 7.6 | 81.0 |
| Perceiver IO++ | UTF-8 bytes | 2048 | 256 | 40 | 425M | 241B | 4.2 | **81.8** |

Table 1: **Perceiver IO on language**: results on the GLUE benchmark (Avg. = average performance, higher is better). Following Devlin et al. (2019) we exclude the WNLI task. We use Pearson correlation on STS-B, Matthews correlation on CoLa and accuracy on the remaining tasks. BERT Base (test) performance is reported from Devlin et al. (2019). SPS = train-time steps per second. $M$ = # inputs and $N$ = # latents.

## 4.1 LANGUAGE

We first compare Perceiver IO to standard Transformers for language. Although Transformers were originally developed for language, their quadratic complexity makes them difficult to use on language inputs without tokenization, which typically shortens the length of input sequences by a factor of $\sim$4. But unlike Transformer-based models such as BERT (Devlin et al., 2019) or XLNet (Yang et al., 2019), Perceiver IO scales linearly with input length. Our experiments focus on showing that Perceiver IO performs as well as or better than Transformers for masked language modeling (MLM) while removing tokenization (which is hard to maintain, introduces engineering overhead, and adds needless complexity to language models (Bostrom & Durrett, 2020; Clark et al., 2022)).

We compare results for a given FLOPs budget rather than a given parameter budget as the former grows quadratically with sequence length but the latter is independent (except for positional encodings). From a practioner's perspective, FLOPs matter more than parameters since FLOPs directly relate to training time. We evaluate the quality of the learned representation on the GLUE benchmark (Wang et al., 2019) and report our results in Tab. 1. We find that at a given FLOPs budget, Perceiver IO trained without tokenization matches the performance of a strong Transformer-based model trained with SentencePiece tokenization (Sennrich et al., 2016; Kudo & Richardson, 2018).

**Pretraining.** We pretrain on the Masked Language Modeling (MLM) task proposed in Devlin et al. (2019) using a large text corpus obtained by combining English Wikipedia and C4 (Raffel et al., 2020). For both the SentencePiece and the byte-level models, we mask 15% of the words, where a word is defined as a space-delimited sequence of characters. As a token contains many bytes on average, we need to increase the sequence length to input a similar amount of text: we use input sequence lengths of 512 SentencePiece tokens or 2048 UTF-8 bytes. For the SentencePiece models we use a vocabulary size of 32,000 following Devlin et al. (2019). For the byte-level models, the vocabulary size is much smaller: 256 bytes and 4 special tokens (`[PAD]`, `[MASK]`, `[CLS]`, `[SEP]`). Perceiver IO produces one output vector per masked input by using learnable position-dependent vectors to query the output of the final latent processing layer. We then apply a position-wise linear layer on top of these output vectors and train the model using a softmax cross-entropy loss to predict the original non-masked input as target. The full details of the architecture are given in Sec. F.2. See Appendix Fig. 7 for analysis and visualization of the learnt features.

**Finetuning.** We finetune Perceiver IO on the GLUE Benchmark Wang et al. (2019), reporting the best performance on the dev set for a fixed size sweep of finetuning hyperparameters. Individual task results and hyperparameters are given in Sec. F.4.

**Perceiver IO on SentencePiece tokens.** We first observe that Perceiver IO applied on SentencePiece tokenized input sequences slightly outperforms a strong BERT baseline applied on the same inputs (81.2 vs 81.1). As a result of the reduced latent size of 256 we can train a much deeper network with 26 processing layers compared to BERT Base (12 layers) while maintaining a similar FLOPs budget.

**Perceiver IO on UTF-8 bytes.** Next, we show that we can leverage Perceiver IO to run on much longer sequences than a regular Transformer. Rather than using a fixed, handcrafted vocabulary, our model works directly with the raw byte inputs: we simply feed in and predict the UTF-8 bytes of the input string. Perceiver IO significantly outperforms a byte-level BERT baseline at the same FLOPs

budget, demonstrating the real advantage of Perceiver IO architecture for language.[1] Remarkably, the bytes Perceiver IO is on par with BERT running on SentencePiece tokens, showing that Perceiver IO is also competitive against strong baselines relying on handcrafted tokenizers. The performance of Perceiver IO on bytes scales well with more FLOPs where we obtain 81.8 on the GLUE benchmark.

The byte-level Perceiver IO shares some similarities with the concurrent CANINE work (Clark et al., 2022). While Clark et al. (2022) rely on a relatively sophisticated pipeline that maps Unicode codepoints to hash embeddings (Svenstrup et al., 2017), we embed raw UTF-8 bytes directly. Clark et al. (2022) also uses a bottleneck architecture to scale to longer text inputs, but their upsampling strategy differs from ours: they concatenate raw inputs with their aligned downsampled latent representation, apply a 1D convolution and then run a shallow transformer stack on the resulting upsampled sequence. Their approach scales quadratically with respect to the original input length while Perceiver IO's decoder scales *linearly* with respect to the target output size. Our work scales to byte-level inputs without making any assumptions about the structure of the input, which allows it to be used beyond language as shown in the following sections.

**Multitask Perceiver IO.** We use multitask queries as described in Sec. 3.2 to finetune on all 8 GLUE tasks simultaneously using the UTF-8 byte model (results in Tab. 2). We compare to results from the single task regime where the model is trained independently on each task. We also compare to an approach analogous to BERT's `[CLS]` token that prepends a special token to the input and uses the position corresponding to this token to query the task logits. We do this either by sharing a single token among tasks (*Shared input token*) or using task-specific tokens (*Task-specific input token*). In both cases, we use a 2-layer task-specific MLP head to generate output logits for each task. We observe that our multitask approach outperforms single-task approaches and matches the approach that uses 8 task-specific input tokens. Our approach is more generic as it decouples the output array from the input array by not relying on `[CLS]` tokens. This is especially appealing when the tasks are many or inhomogeneous, as we show in Sec. 4.3.

| Method | Avg. |
|---|---|
| Single-task query | 81.0 |
| *Multitask* | |
| Shared input token | 81.5 |
| Task-specific input tokens | **81.8** |
| Multitask query | **81.8** |

Table 2: Multitask Perceiver IO. Results use the same metric as Tab. 1 (higher is better).

## 4.2 OPTICAL FLOW

Optical flow is a decades-old open problem in computer vision (Lucas & Kanade, 1981; Horn & Schunck, 1981). Given two images of the same scene (e.g. two consecutive frames of a video), the task is to estimate the 2D displacement for each pixel in the first image. This has many broader applications, such as navigation and visual odometry in robots (Campbell et al., 2004), estimation of 3D geometry (Ranftl et al., 2020), and even to aid transfer of more complex, learned inference such as 3D human pose estimation from synthetic to real images (Doersch & Zisserman, 2019). Optical flow is challenging for neural networks for two reasons. First, optical flow relies on finding correspondence: a single frame provides no information about flow, and images with extremely different appearance can produce the same flow. Second, flow is extremely difficult to annotate, and the few datasets with realistic images and high-quality ground truth are small and biased. While it is straightforward to generate large synthetic datasets as training data, e.g. AutoFlow (Sun et al., 2021), there is still a large domain gap.

Algorithms for optical flow thus must learn to accomplish several steps in a way that transfers from synthetic to real data. First, the algorithm must find correspondence between points. Then it must compute their relative offsets. Finally it must propagate flow across large regions of space, including to parts of the image which have no texture for correspondence. To generalize to real data, the learned procedure needs to work for objects and textures that weren't seen in the training data.

These difficulties have led flow researchers to develop some of the most involved architectures in the computer vision literature. State of the art algorithms, such as PWCNet (Sun et al., 2018), RAFT (Teed & Deng, 2020) or GMA (Jiang et al., 2021), use explicit machinery to ensure each of these steps is performed correctly even on out-of-domain data. Expensive global correlation

---

[1]Despite its greater depth, Perceiver IO is also faster than the Transformer-based BERT baselines in real wall-clock terms – by over a factor of 2 for the byte-based models – as shown in Tab. 1.

volumes explicitly compare features within a spatiotemporal neighborhood across images to find correspondences. Flows are computed iteratively and hierarchically in 2D space using explicit lookup operators to verify correctness, leading to slow performance on TPUs (Jouppi et al., 2017).

**Perceiver IO on Flow** In contrast, we apply Perceiver IO to flow in a straightforward manner. We concatenate the frames along the channel dimension and extract a $3 \times 3$ patch around each pixel (leading to $3 \times 3 \times 3 \times 2 = 54$ values for each pixel). We concatenate a fixed position encoding to these features and then apply Perceiver IO. To decode, we query the latent representation using the input encoding. See Sec. H for training details and results with various forms of pre- and post-processing, which typically perform similarly. We also test a version with convolutional downsampling and RAFT-style upsampling, which performs only slightly worse while improving computation time.

It may seem counter-intuitive to append the images along the channel dimension, as large motions might result in pixels on entirely different objects being concatenated. However, this kind of operation isn't unprecedented: one of the earliest optical flow algorithms, Lucas-Kanade (Lucas & Kanade, 1981), makes explicit use of the *temporal* image gradient, which is approximated by the difference in intensities at a given pixel across two frames. The algorithm uses the fact that the temporal gradient of the image approximates the spatial gradient times the spatial velocity, if lighting effects are ignored. The approximation is even better for image regions with very little texture. Such regions are challenging for algorithms that attempt to find explicit correspondence in feature space, especially if feature encoding involves any normalization operations, which may destroy intensity information.

**Results** Tab. 3 shows our results, following the standard protocol for training on AutoFlow (Sun et al., 2021). We compare to PWCNet and RAFT baselines trained by the AutoFlow authors. On Sintel (Butler et al., 2012), our results are slightly better than RAFT on Sintel and outperform PWC-Net on KITTI (Menze & Geiger, 2015). As far as we are aware, this result is state of the art on Sintel.final (GMA Jiang et al. (2021)

| Network | Sintel.clean | Sintel.final | KITTI |
|---|---|---|---|
| PWCNet (Sun et al., 2018) | 2.17 | 2.91 | 5.76 |
| RAFT (Teed & Deng, 2020) | 1.95 | 2.57 | **4.23** |
| Perceiver IO | **1.81** | **2.42** | 4.98 |

Table 3: Optical Flow evaluated on Sintel (Butler et al., 2012) and KITTI with average end-point error (EPE) (lower is better). Baselines are reported from Sun et al. (2021).

produces slightly better numbers on the somewhat easier Sintel.clean evaluation set using different training data). This is surprising considering how different our architecture is from PWCNet and RAFT and how little tuning for flow Perceiver IO required. We use no cost volumes or explicit warping, our model is not explicitly hierarchical, and the latent representation doesn't even maintain the 2D layout of the inputs. Also note that we reuse RAFT's AutoFlow augmentation parameters, which were tuned specifically for RAFT using population-based training (Sun et al., 2021). As shown in Appendix Fig. 8, qualitatively Perceiver IO is good at following object boundaries, and can easily propagate motion across image regions with little texture.

### 4.3 MULTIMODAL AUTOENCODING

We explore using Perceiver IO for audio-video-label multimodal autoencoding on the Kinetics-700-2020 dataset (Smaira et al., 2020). The goal of multimodal autoencoding is to learn a model that can accurately reconstruct multimodal inputs in the the presence of a bottleneck induced by an architecture. This problem has been previously studied using techniques such as Restricted Boltzmann Machines (Ngiam et al., 2011), but on much more stereotyped and smaller scale data.

Kinetics-700-2020 has video, audio, and class labels. We wish to train a model to reconstruct all modalities simultaneously. With traditional autoencoding models like convolutional encoder-decoders, it is not obvious how to combine these modalities, because each uses data of different dimensions – 3D (video), 1D (raw audio), and 0D (class labels) – and with wildly different numbers of elements. With Perceiver IO, we pad the inputs with modality-specific embeddings, serialize them into a single 2D input array and query outputs using queries containing position encodings (for video and audio) and modality embeddings.

We train on 16 frames at $224 \times 224$ resolution, preprocessed into 50k 4x4 patches as well as 30k raw audio samples, producing a total of 1920 16-d vectors and one 700-d one-hot class label. We decode directly into pixels, raw audio, and the one-hot label without any post-processing. To prevent the

Figure 4: Multimodal audio-video-label autoencoding with 88x compression. Side-by-side: inputs on left, reconstructions right. See the supplemental material for example output video and audio.

model from encoding the label directly into one of the latent variables, we mask the class label 50% of the time in training. Due to the scale of inputs and outputs in this task we subsample decoding in training, while fully decoding in testing: we sampled 512 audio samples and 512 pixels and the class label for every training example. This allows us to directly decode to a video-sized array, which would otherwise be infeasible given memory constraints. We used a latent array with 512 channels and 784, 392, and 196 latents, resulting in compression ratios of 88x, 176x, and 352x respectively.

| Compression Ratio | Audio PSNR | Video PSNR | Top-1 Accuracy |
|---|---|---|---|
| 88x | 26.97 | 24.37 | 10.2% |
| 176x | 25.33 | 24.27 | 8.6% |
| 352x | 14.15 | 23.21 | 11.5% |

Table 4: Multimodal autoencoding results. Higher is better for accuracy and PSNR.

We show results in Tab. 4 and reconstructions in Fig. 4. By masking the classification label during evaluation, our autoencoding model becomes a Kinetics 700 classifier. Latent variables are shared across modalities, so the quality of reconstructions for each modality is sensitive to the weight of its loss term and other training hyperparameters. Tab. 4 shows one tradeoff, where we emphasized video and audio PSNR at the expense of classification accuracy. By putting stronger weight on the class loss, we can reach 45% top-1 accuracy while maintaining 20.7 PSNR for video (Sec. I). This strongly suggests that Perceiver IO can jointly represent modalities with very different properties.

### 4.4 IMAGENET, STARCRAFT II, AND AUDIOSET

**Please read the Appendix for results on ImageNet (Sec. A), StarCraft II (Sec. B), and AudioSet (Sec. C)**. We have omitted these results from the main paper to make the exposition as clear as possible within 9 pages (the ICLR camera ready page limit). As highlights of these experiments: (1) on ImageNet, Perceiver IO surpasses 80% top-1 accuracy (84.5% top-1) without using 2D convolutions after pretraining on JFT. (2) When used to replace AlphaStar's entity Transformer, Perceiver IO obtains a $\sim 3.5\times$ reduction in FLOPs while preserving StarCraft II 87 % win rate and parameter count, after only 3 experimental runs. (3) On AudioSet, Perceiver IO consistently outperforms the original Perceiver when using the same training protocol on multimodal video + audio classification. The Appendix includes additional details of the experimental domains included in the main paper.

## 5 CONCLUSION

In this work we introduce Perceiver IO, an architecture capable of handling general purpose inputs and outputs while scaling linearly in both input and output sizes. As we show, this architecture achieves good results in a wide variety of settings, making it a promising candidate for a *general purpose* neural network architecture. Perceiver IO leverages the expressive power of latent attention and uses learned queries to expose a simple and unified interface that can handle multimodal and multitask settings. Overall, Perceiver IO offers a promising way to simplify the construction of sophisticated neural pipelines and facilitate progress on multimodal and multiask problems.

### ACKNOWLEDGMENTS

We are grateful to Ankush Gupta and Adrià Recasens Continente for reviewing drafts of this paper and to Deqing Sun for sharing code and helpful advice on the optical flow experiments.

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

## APPENDIX

| Modalities | Tasks | Preprocessing | Postprocessing | # Inputs | # Outputs |
|---|---|---|---|---|---|
| Text | Token-level pred. | Tokenization + Embed. | Linear projection | $512 \times 768$ | $512 \times 768$ |
| Text | Byte-level pred. | Embed. | None | $2,048 \times 768$ | $2,048 \times 768$ |
| Text | Multi-task (8 tasks) | Embed. | None | $2,048 \times 768$ | $8 \times 768$ |
| Video | Flow prediction | None | None | $365,056 \times 64$ | $182,528 \times 64$ |
| Video | Flow prediction | Concat | None | $182,528 \times 64$ | $182,528 \times 64$ |
| Video | Flow prediction | Conv+maxpool | RAFT upsampling | $22,816 \times 64$ | $11,408 \times 64$ |
| Video | Flow prediction | Conv+maxpool+concat | RAFT upsampling | $11,408 \times 64$ | $11,408 \times 64$ |
| Video+Audio+Label | Autoencoding | Patch: 1x4x4 Vid, 16 Aud | None | $50,657 \times 704$ | $803,297 \times 512$ |
| Image | Classification | None | None | $50,176 \times 3$ | $1 \times 1,000$ |
| Image | Classification | Linear projection | None | $50,176 \times 256$ | $1 \times 1,000$ |
| Image | Classification | Conv+maxpool | None | $3,136 \times 64$ | $1 \times 1,000$ |
| StarCraft Unit Set | Encoding and Classification | Tokenization | Pointer network | $512 \times 256$ | $512 \times 128$ |
| Video+Audio | Classification | Patch: $2 \times 8 \times 8$ Vid, 128 Aud | None | $13,024 \times 487$ | $1 \times 527$ |
| Video+Audio | Classification | Patch: $2 \times 8 \times 8$ Vid. Aud $\to$ mel-spectrogram | None | $17,344 \times 487$ | $1 \times 527$ |

Table 5: Details of each of the tasks we use to evaluate Perceiver IO here. The positional and task embeddings appended to inputs for each case are listed in Tab. 6.

In the following sections, we describe experiments on three additional domains (ImageNet, StarCraft II, and AudioSet) and provide additional details for the methods and experiments described in the paper. For ease of reference and comparison across domains, we describe the input and output size and processing used in all experiments in Tab. 5 and provide details of input key/value, position encoding, and output queries used in all experiments in Tab. 6.

On all domains but StarCraft II, we include experiments with several input configurations, ranging from no domain adaptation (e.g. tokenizer-free language, flow from raw pixels, ImageNet with no convolutional or patch-based preprocessing and fully learned position encodings) to moderate domain adaptation (e.g. SentencePiece language understanding, flow from conv+maxpool-processed images and with RAFT upsampling, ImageNet with conv+maxpool-preprocessing and 2D Fourier features). These results demonstrate the unprecedented generality of Perceiver IO, the simplicity that this architecture unlocks in handling a range of tasks, and its flexibility to work as part of a domain-adapted system.

| Domain | Input Modality | Encoder KV input | Encoder KV channels | Decoder query input | Decoder query channels |
|---|---|---|---|---|---|
| Language (MLM) | Text | byte/token encoding + learned pos | 768 | learned pos | 1280 |
| Language (Perceiver IO++ MLM) | Text | byte/token encoding + learned pos | 768 | learned pos | 1536 |
| Language (GLUE) | Text | byte/token encoding + learned pos | 768 | Class query (per-task) | 1280 |
| Language (Perceiver IO++ GLUE) | Text | byte/token encoding + learned pos | 768 | Class query (per-task) | 1536 |
| Optical Flow | Video (concat. frames) | [conv or Linear(concat RGB), 2D FFs] | 322 | [Linear(RGB), 2D FFs] | 322 |
| Optical Flow | Video | [conv or Linear(RGB), 3D FFs] | 451 | [conv features, 3D FFs] | 451 |
| Kinetics | Video, | [patched RGB, 3D FFs, learned modality feat.] | 704 | [3D FFs, learned modality feat.] | 1026 |
| | Audio, | [patched sound pressure, 1D FF, learned modality feat.] | 704 | [1D FF, learned modality feat.] | 1026 |
| | Label | [one-hot label, learned modality feat.] | 704 | [learned modality feat.] | 1026 |
| ImageNet (2D FFs) | Image | [RGB, 2D FFs] | 261 | Class query (single) | 1024 |
| ImageNet (learned pos) | Image | [Linear(RGB), learned pos] | 512 | Class query (single) | 1024 |
| ImageNet (conv) | Image | [Conv features, 2D FFs] | 322 | Class query (single) | 1024 |
| StarCraft II | SC2 entities | Entity features | 128 | Entity features | 128 |
| AudioSet | Video, | [patched RGB, 3D FFs, learned modality feature] | 487 | Class query (single) | 1024 |
| | Audio | [patched sound pressure, 1D FFs, learned modality feature] | 487 | | |
| AudioSet | Video, | [patched RGB, 3D FFs, learned modality feature] | 487 | Class query (single) | 1024 |
| | Mel-spectrogram | [mel-spectrogram features, 1D FFs, learned modality feature] | 487 | | |

Table 6: **Table best viewed on a screen.** The structure and size of the positional and task embeddings used to construct Perceiver IO's encoder key-value inputs and decoder query inputs, for each domain described in the main text. "[x, y]" indicates that x's and y's features are concatenated, while "x + y" indicates that x's and y's features are added to produce the full featurization. "FF" = Fourier features as in Jaegle et al. (2021).

## A  IMAGE CLASSIFICATION

Perceiver did well on ImageNet (Deng et al., 2009) classification without using 2D structure in the design of the architecture, but generated class scores using a simple average + project decoder (see Sec. E.3 and Fig. 6 for a diagram illustrating the difference between the two forms of decoder). We now evaluate the effect of this more general decoder. See Sec. C for similar validation on AudioSet.

| Model | Pretrained? | Accuracy | FLOPs | Params |
|---|---|---|---|---|
| **ConvNet baselines** | | | | |
| ResNet-50 (He et al., 2016) | N | 78.6 | 4.1B | 26M |
| NFNet-F6+SAM (Brock et al., 2021) | N | 86.5 | 377.3B | 438.4M |
| Meta Pseudo Labels (Pham et al., 2021) | Y | 90.2 | - | 480M |
| **ViT baselines** | | | | |
| ViT-B/16 (Dosovitskiy et al., 2021) | N | 77.9 | 55.4B | 86M |
| ViT-H/14 (Dosovitskiy et al., 2021) | Y | 88.6 | - | 632M |
| DeiT 1000 epochs (Touvron et al., 2021a) | N | 85.2 | - | 87M |
| CaiT-M48 448 (Touvron et al., 2021b) | N | 86.5 | 329.6B | 356M |
| **w/ 2D Fourier features** | | | | |
| Perceiver | N | 78.6 | 404B | 42.1M |
| Perceiver IO, config A | N | 79.0 | 407B | 48.4M |
| Perceiver IO, config B (pretrained) | Y | 84.5 | 213B | 212M |
| **w/ learned position features** | | | | |
| Perceiver (learned pos) | N | 67.6 | 404B | 55.9M |
| Perceiver IO, config A (learned pos) | N | 72.7 | 407B | 62.3M |
| **w/ 2D conv + maxpool preprocessing** | | | | |
| Perceiver (conv) | N | 77.4 | 367B | 42.1M |
| Perceiver IO, config A (conv) | N | 82.1 | 369B | 48.6M |
| Perceiver IO, config B (conv) (pretrained) | Y | 86.4 | 176B | 212M |

Table 7: Results on ImageNet image classification (top-1 accuracy, higher is better). "-" indicates a value we could not find reported in the literature. We did not extensively tune our models for efficiency on image classification – the primary focus of this work is generality, rather than speed on images – Perceiver IO uses comparable FLOPs to attention-based image classification models, especially for the more compact configuration B pretrained on JFT. The positional encoding does not significantly change model FLOPs.

**Results** Tab. 7 shows our results alongside representative numbers from the literature. Perceiver and Perceiver IO differ in their decoder, and neither model uses convolutional preprocessing by default. Perceiver IO consistently outperforms the original architecture. After pretraining on JFT (Sun et al., 2017), Perceiver IO performs in the ballpark of models designed primarily for image classification. Perceiver IO is competitive with members of the Vision Transformer (ViT) (Dosovitskiy et al., 2021) family even without relying on 2D convolutions. Perceiver IO is also compatible with convolutional preprocessing: adding a 2D conv+maxpool preprocessing stage leads to a moderate increase in efficiency and bump in performance.

While neither the Perceiver and Perceiver IO incorporate any 2D spatial structure architecturally, they use positional features that inject 2D spatial information (Sec. 3.2 and Appendix sec. D of Jaegle et al. 2021). By replacing these 2D position features with a fully learned position encoding as used on language, we can learn an image classification model that is given no privileged information about the structure of images. This positional encoding is an array of shape $50,176 \times 256$, which is randomly initialized using a truncated Gaussian distribution with scale 0.02. ImageNet networks that use this positional encoding are given no information about 2D image structure. For these experiments, we additionally use a 1D convolutional network to project the RGB at each point to 256 before concatenating it with the learned positional encoding. The results of this experiment are shown in Tab. 7 (**w/ learned position features**). To our knowledge, this is the best result by any model on ImageNet without 2D architectural or feature information.

## A.1 DETAILS OF IMAGENET TRAINING

For ImageNet experiments, we use CutMix (Yun et al., 2019) and MixUp (Zhang et al., 2018) regularization, in addition to RandAugment (Cubuk et al., 2020) as used in Jaegle et al. (2021). We observed only marginal improvements in performance from this change, but it brings the augmentation strategy more in line with the strategy used elsewhere in the literature (Brock et al., 2021; Touvron et al., 2021a). In all experiments, we use RandAugment with 4 layers at magnitude 5 (as in Jaegle

| Model | Train steps/sec |
|---|---|
| Perceiver (2D FF) | 4.73 |
| Perceiver IO (2D FF) | 4.85 |
| Perceiver (learned pos) | 4.16 |
| Perceiver IO (learned pos) | 4.14 |
| Perceiver (conv) | 4.73 |
| Perceiver IO (conv) | 5.58 |
| Perceiver IO (pretrained) | 6.41 |

Table 8: ImageNet model training speed. The model used for pretraining is faster because it uses only 16 process modules. We did not reimplement baselines, so we report only the training speed of Perceiver and Perceiver IO models.

et al. 2021) and CutMix with a ratio of 0.2. In early experiments, we found that higher weight decay and moderate gradient clipping contributed to better generalization: we use a weight decay of 0.1 and clip to a maximum global gradient norm of 10. We use no dropout. We use an architecture with weight sharing in depth: the latent (processing) component of the architecture includes 8 blocks of 6 attention modules each, and weights are shared between the corresponding modules in each block. We omit the repeated encoder cross-attends used in Jaegle et al. (2021) as we found these to lead to relatively small performance improvements but to significantly slow down training: using 8 encoder cross-attention instead of 1 adds an additional 303 billion FLOPs. The FLOPs for all ImageNet models presented here are given in Tab. 7 and the model training step time on 64 TPUv3 are given in Tab. 8.

For all ImageNet experiments, we train for 110 epochs, using a batch size of 1024 and 64 TPUs. We use LAMB with a simple learning rate schedule consisting of a flat learning rate of $2 \times 10^{-3}$ for 55 epochs, after which the learning rate is decayed to 0 over the final 55 epochs following a cosine decay schedule (Loshchilov & Hutter, 2017). We found a cosine learning rate decay schedule simpler to tune than the step decay schedule used in Jaegle et al. (2021) and that beginning the decay process halfway through training generally led to good performance without introducing instability. We found it important to omit an initial learning rate warm-up period, as this often prevented models from training when using LAMB.

## A.2 LARGE-SCALE PRETRAINING

As reported in Jaegle et al. (2021), Perceiver models are able to easily overfit ImageNet-scale datasets without regularization. For this reason, we explored pretraining a model on JFT, a large-scale, multi-labeled internal dataset with 300 million images spanning approximately 18,000 classes (Sun et al., 2017). We pretrain on this dataset at the same resolution used on ImageNet ($224 \times 224$) using a base learning rate of $3 \times 10^{-4}$ and a cosine decay schedule, decaying to 0 over 14 epochs. We omit all augmentation except basic cropping, resizing, and left-right flipping. We use a weight decay of 0.1. We use a larger batch size of 8192 and train on 256 TPUs. Images in this dataset come with a variable number of labels, so we use a cross-entropy loss with a multi-one-hot representation of the targets. Unlike in the other ImageNet experiments, we do not share weights in the latent self-attention process modules, but use a 16-layer latent network with no weight sharing in depth. Unlike the other ImageNet experiments, the process-module MLPs use a hidden layer with $4\times$ the number of channels (rather than $1\times$ as on other ImageNet experiments). When pretraining the 2D FF model, we use a 1D convolutional network to project input RGB at each point to 256 before concatenating it with the positional encoding (a 2D Fourier frequency positional encoding). When pretraining the conv+maxpool model, we instead use the initial convolutional preprocessing described in Sec. A.3 below.

To evaluate transfer, we fine-tune our pre-trained model on ImageNet. We replace only the final linear layer of the decoder to produce the required 18,000 classes. For 2D FF fine-tuning, we used similar optimizer and augmentation settings as with our from-scratch ImageNet training: 1024 batch size on 64 TPUs, 131K steps with LAMB using a flat base LR of 0.002 for the first 70K steps and a cosine

| Entity encoder | Win rate | Params (M) | FLOPs | Train steps/sec |
|---|---|---|---|---|
| Transformer (Vinyals et al., 2019) | 0.87 | 144 | 3.3B | 2.9 |
| Perceiver IO | 0.87 | 140 | 0.93B | 2.9 |

Table 9: We evaluate Perceiver IO on StarCraft II by using it to replace the well-tuned Transformer entity encoder. Perceiver IO matches the performance of the original Transformer despite using fewer FLOPs and parameters and requiring essentially no tuning. Note that the training steps/sec of the overall system does not change because the entity encoder is not the speed bottleneck.

learning rate decay for the last 61K steps. We use identical settings for conv+maxpool fine-tuning with the exception of the base learning rate, which we set to 0.0002, as training with the higher 0.002 rate was unstable.

### A.3   2D CONVOLUTIONAL PREPROCESSING ON IMAGENET

In other image settings discussed here, we optionally use simple pre- and post-processing steps to reduce the size of very large inputs and outputs. Because ImageNet data points are relatively small (Tab. 5), we are able to process full images without convolutional pre- and post-processing. Consequently, we can use this dataset to probe the sensitivity of the model to convolutional pre-processing. Incorporating a single convolution + max pooling leads to a moderate improvement in the performance of the architecture: this is perhaps unsurprising, as convolutional pre-processing injects information about the 2D structure of images into the architecture. By comparison ViT first processes images by applying a 2D convolution with matched kernel and stride to downsample its inputs (referred to as a "linear projection of flattened patches" in that work and throughout the ViT literature). As in other experiments, we find that incorporating an attention-based decoder (Perceiver IO) leads to better results than averaging and pooling the output (Perceiver). Using convolutional preprocessing leads to a moderate reduction in the number of FLOPs used by the model (Tab. 7) and training speed in some configurations (Tab. 8). The input to the network after preprocessing is $56 \times 56$ instead of $224 \times 224$ as in the experiments directly on pixels.

## B   STARCRAFT II

To further demonstrate Perceiver IO's capabilities on discrete modalities and as a drop-in replacement for Transformers, we plug in Perceiver IO in place of AlphaStar's Transformer. AlphaStar (Vinyals et al., 2019) is the state-of-the-art system for the challenging real-time strategy game of StarCraft II.

At its core, AlphaStar represents the units in the game as a discrete, unordered set of symbols (the "units"). These units are represented by a vector of properties including unit type, position, and health. At each timestep, the architecture encodes units with an entity encoder, which in the original model was parameterized using a vanilla Transformer.

The entity encoder takes as input a set of 512 entities (referred to as `embedded_entity` in Vinyals et al. (2019)) and produces as output an embedding for each entity (`entity_embeddings`) and a 1D embedding reduced over entities (`embedded_entity`). These 512 entities represent the units and other entities that are present in the game: unused entity slots are masked. `entity_embeddings` is produced by passing the outputs of the entity encoder through a ReLU and a 1D convolution with 256 channels. `embedded_entity` is produced by averaging the (unmasked) entity encoder outputs and passing it through a linear layer with 256 units and a ReLU.

In the original AlphaStar system, the entity encoder consisted of a Transformer with 3 attention layers, each of which used 2 heads and a feature dimension of 128. The output of each attention layer is projected to 256 and followed by an 2-layer MLP with hidden size 1024 and output size 256. This architecture was arrived by an extensive tuning process as reported in Vinyals et al. (2019).

The representation produced by the entity encoder is used both as a summary of the state (after pooling) and as a rich representation of the units. This representation is used by a pointer network (Vinyals et al., 2015) to assign a probability to each possible unit selection, in the process parameterizing the agent's unit selection policy. For this reason, we view AlphaStar as an important test case for Perceiver IO's ability to function as a general-purpose tool for processing symbolic or set-valued

| Model | Input | mAP | Latent channels ($D$) | Params (M) | FLOPs | Train steps/sec |
|-------|-------|-----|------------------------|------------|-------|-----------------|
| Perceiver | Raw audio + video | 42.4 | 512 | 21.0 | 52.3B | 3.8 |
| Perceiver IO | Raw audio + video | 43.3 | 512 | 25.0 | 52.9B | 3.8 |
| Perceiver | mel-spectrogram + video | 43.6 | 512 | 21.0 | 60.7B | 3.8 |
| Perceiver IO | mel-spectrogram + video | 44.9 | 1024 | 88.2 | 129.5B | 3.8 |

Table 10: Perceiver IO on multimodal (audio + video) AudioSet classification (mAP = mean average precision, higher is better). All models have similar runtimes despite FLOPs differences because the bottleneck is data loading and preprocessing rather than model forward/backward passes.

data. If the question is "can Perceiver IO serve as a replacement for a well-tuned Transformer as a symbolic processing engine?" then the answer is yes:

We obtained StarCraft II results by using Perceiver IO instead of a Transformer for the AlphaStar entity encoder. We replaced the Transformer with a Perceiver IO with a latent of index dimension 32, keeping the input and output size of 512 units. We performed **no** tuning beyond sweeping the size of the latent index dimension (we tried values of 32 and 64): Perceiver IO works out of the box. We observed that the resulting agent reached the same level of performance as the original AlphaStar agent, reaching an 87% win-rate versus the Elite bot after behavioral cloning (Pomerleau, 1989) on human data, while also leading to a $3\times$ decrease in FLOPs (Tab. 9).

We replaced this Transformer with a 3-layer Perceiver IO with a latent of index dimension 32. We tuned only the size of the index dimension (sweeping values of 32 and 64), but otherwise used the same hyperparameters as ImageNet.

## C  AUDIOSET

We seek to confirm that the the attention-based decoder helps even on classification, where the original Perceiver's decoder could be used. We show that the trend identified on ImageNet holds more generally, by revisiting the multimodal AudioSet classification domain. AudioSet is a large-scale event classification dataset containing 1.7 million training examples, each consisting of 10s long video and audio. Each example is labeled with several labels drawn from 527 classes.

We perform experiments using the protocol described in Jaegle et al. (2021), training models for 100 epochs using 32-frame clips at train time and 16 overlapping 32-frame clips at test time. As in the ImageNet experiments, We compare the performance of Perceiver and Perceiver IO using models that are matched except for the decoder (we use an average + project decoder for Perceiver and a query-based attention decoder for Perceiver IO, see Sec. E.3 and Fig. 6). All models use an architecture with 12 processor modules and a latent index dimension $N$ of 512 (we omit the repeated cross-attends used in Jaegle et al. (2021)). We compare models taking video and either raw audio or mel-spectrogram (pre-processed audio) as input. For all four model settings, we swept the number of latent channels (using $D \in \{512, 1024\}$) and report the best value for each setting. We performed no additional tuning.

Results of this experiment are shown in Tab. 10. We find that as in the ImageNet experiments, using the attention-based decoder leads to small but consistent improvements over the less generally applicable average + project decoder. Because Perceiver IO introduces no domain assumptions not present in the original Perceiver, this is evidence that Perceiver IO is a strictly more general model.

## D  FLOPS CALCULATION

In all cases, we report theoretical FLOPs with multiplies and accumulates counted as separate operations. This is the strategy used in Kaplan et al. (2020) and elsewhere in the literature. We use this strategy consistently here to allow comparisons between the models we propose and develop (including our BERT reimplementation). Note that some papers in the literature report FLOPs using fused multiply-accumulates: using this strategy will cut the figures reported here in half.

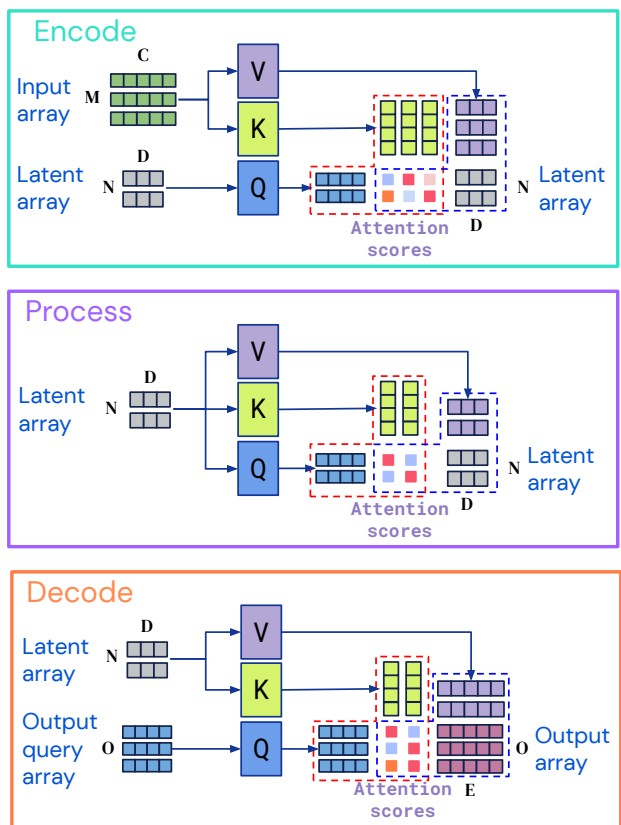

Figure 5: Schematic depiction of encode, process, and decode attention. Each attention module uses the same operations, but differs in which inputs are used to generate key/values or queries and in the output shape. Encode attention can be viewed as mapping an input to a latent space, typically with a smaller index dimension (fewer elements). Decode attention can be viewed as mapping a latent to an output space, often with a larger index dimension (more elements). Both of these are forms of cross-attention. Process attention (self-attention) preserves the input index dimension (same elements). Red and blue dashed lines are used to highlight the two matrix multiplications used in QKV attention, as described in the text.

# E  ARCHITECTURAL DETAILS

Perceiver IO is constructed from GPT-2-style (Radford et al., 2019) Transformer attention modules, which consist of QKV attention followed by an MLP, along with linear projection layers to ensure inputs to and outputs from the QKV attention and MLP take on desired sizes. Using the array sizes of the encoder attention, the QKV attention takes in two two-dimensional arrays, a key-value input array $X_{KV} \in \mathbb{R}^{M \times C}$ and a query input array $X_Q \in \mathbb{R}^{N \times D}$, and maps them to an array $X_{QKV} \in \mathbb{R}^{N \times D}$, sharing the shape of the query input (after projection). $X_{QKV}$ is used as input to an MLP, which is applied independently to each element of the index dimension (i.e. convolving the MLP with its input along the first dimension), producing a final array $X_{MLP} \in \mathbb{R}^{N \times D}$.

While we describe attention as taking two inputs, in standard Transformers it is typically described as mapping one input to an output of the same size. This is because all modules of a standard Transformer use *self*-attention, where the same input is used for both key-value inputs and query inputs. The view of attention that we describe encompasses both cross-attention and self-attention, both of which are specific ways of using QKV-attention. Perceiver IO uses cross-attention for encoder and decoder attention modules and uses self-attention for the latent processing modules. These modules differ primarily in terms of what shape data they ingest and produce (Fig. 5).

We now describe the structure of QKV attention and the MLP in more detail.

### E.1 ATTENTION MODULE INTERNALS

QKV attention takes in two two-dimensional arrays, a query input $X_Q \in \mathbb{R}^{N \times D}$ and a key-value input $X_{KV} \in \mathbb{R}^{M \times C}$. The output of QKV attention is an array with the same index (first) dimension as the query input and a channel (second) dimension determined by an output projection:

$$Q = f_Q(X_Q); \, K = f_K(X_{KV}); \, V = f_V(X_{KV}) \tag{1}$$
$$X_{QK} = \text{softmax}(QK^T/\sqrt{F}) \tag{2}$$
$$\text{Attn}(X_Q, X_{KV}) = X_{QKV} = f_O(X_{QK}V), \tag{3}$$

where $X_{QK}$ is an array of attention maps $\in \mathbb{R}^{N \times M}$, and $X_{QKV}$ is an array $\in \mathbb{R}^{N \times D}$. The functions $f_{\{Q,K,V\}}$ are linear layers mapping each input to a shared feature dimension $F$ and $f_O$ is a linear layer projecting the output to a target channel dimension, which is often the same size as $X_Q$'s. All linear layers are applied convolutionally over the index dimension (the first dimension of their inputs). We have omitted batch and head dimensions (in the case of multi-headed attention) for readability. QKV attention is followed by a two-layer MLP with a GELU (Hendrycks & Gimpel, 2016) nonlinearity following the first layer. The full module has the following structure:

$$X_{QKV} = \text{Attn}(\text{layerNorm}(X_Q), \text{layerNorm}(X_{KV})) \tag{4}$$
$$X_{QKV} = X_{QKV} + X_Q \tag{5}$$
$$X_{QKV} = X_{QKV} + \text{MLP}(\text{layerNorm}(X_{QKV})), \tag{6}$$

slightly abusing notation for simplicity and to emphasize the residual structure. "Attn" refers to QKV as described above.

In the context of decoder attention, we sometimes find it helpful to omit the second step ($X_{QKV} = X_{QKV} + X_Q$), as it involves adding the model output with a query. Queries sometimes include features inherited from the input space (Tab. 6), and this residual connection may make learning unnecessarily difficult. For example, for optical flow, including this residual connection forces the network to produce optical flow output by adding RGB and Fourier features to the model's output.

### E.2 COMPUTATIONAL COMPLEXITY

The computational complexity of each attention module is dominated by the two matrix multiplications in QKV attention. Still using the shapes of the encoder attention, these two matrix multiplies involve matrices of shape $M \times F$ and $N \times F$ and $M \times N$ and $N \times F$, giving overall time and memory complexity of $\mathcal{O}(MNF)$. Let $M$, $N$, and $O$ be the index dimensions for the input, latent, and output arrays, and to simplify the analysis let $F$ be the feature size for all layers. The KV and Q sizes for the encoder, latent transformer, and decoder will then be $M \times F$ and $N \times F$ (for the encoder), $N \times F$ and $N \times F$ (for the latent transformer), and $N \times F$ and $O \times F$ (for the decoder). A model with $L$ latent attention blocks has complexity $\mathcal{O}([M + O + LN]NF)$. In other words, Perceiver IO has complexity linear in the size of the input and output arrays and it decouples the depth of the latent transformer from the input and output sizes. Both of these properties contribute to Perceiver IO's efficiency: while many proposals for efficient attention modules or architectures include linear or sub-quadratic scaling with input/output size, Perceiver IO is unusual in also decoupling depth from input/output size (without requiring domain-specific strategies like 2D convolution). For further discussion of these points, see Sec. 2 and Sec. A of Jaegle et al. (2021).

### E.3 USING THE DECODER FOR CLASSIFICATION / REGRESSION

As we show in ImageNet and AudioSet experiments, the attentional decoder used here can be used in settings where standard average + project decoders are applicable. We find that the attentional decoder typically produces somewhat better results than the standard decoder. This likely occurs because attentional decoding is more expressive than average + project decoding. To make this clear, we illustrate the two pooling schemes in Fig. 6. Both decoders can be viewed as first averaging the

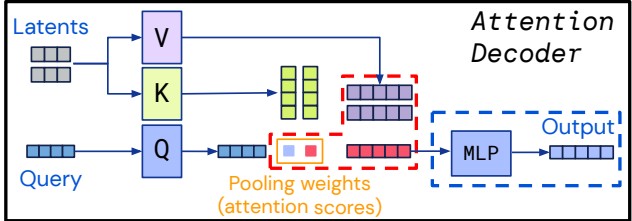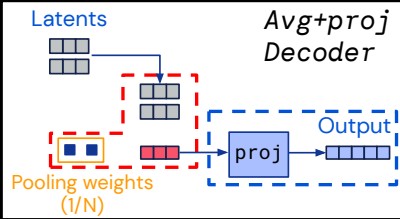

Figure 6: Single-query attention decoder (left), as used in Perceiver IO for classification tasks and a standard average + project decoder (right), as used in Jaegle et al. (2021). Both modules can be seen as first aggregating latents by weighted averaging (learned, data-dependent weighting for the attention decoder; uniform weights for the average + project decoder) and then projecting to an output channel dimension (linear value projection + MLP for the attention decoder; simple linear projection by the average + project decoder). Attentional decoding is more expressive than average + project decoding and follows the same architectural template as encoder and processor modules.

latents and then projecting them to a target shape, but decoder attention uses more expressive modules for each of these operations. Instead of uniformly weighting each input in the averaging operation, decoder attention uses the attention scores as data-dependent weights for each input point. Instead of projecting the raw averaged input to a target dimensionality, decoder attention first projects inputs via a value layer and then processes them with an MLP. In addition to its greater expressivity, decoder attention has the advantage of being easily generalizable to dense outputs (by increasing the number of queries) and of reusing the same architectural pattern used for the encoder and processor modules.

## F LANGUAGE: ADDITIONAL DETAILS

### F.1 OTHER TOKENIZER-FREE MODELS

One application of Perceiver IO is byte-level language processing, which has concurrently been addressed by several other groups. Clark et al. (2022) trains models on Unicode code points and shows results competitive with subword-based models on a multilingual question answering dataset. Tay et al. (2022) trains on UTF-8 bytes directly by introducing a hand-designed module that is trained end-to-end to perform subword tokenization and produces results on-par with and sometimes better than subword-based models. Xue et al. (2022) trains encoder-decoder T5 models on UTF-8 bytes directly and shows that making the encoder 3x deeper than the decoder leads to comparable performance with subword baselines.

### F.2 ARCHITECTURE DETAILS

The architecture hyperparameters and the training speed for the Perceiver IO used in the language experiments are given in Tab. 11.

| Model | BERT Base | BERT matching FLOPs | Perceiver IO Base | Perceiver IO | Perceiver IO++ |
|---|---|---|---|---|---|
| Tokenizer | SentencePiece | UTF-8 bytes | SentencePiece | UTF-8 bytes | UTF-8 bytes |
| Number of inputs ($M$) | 512 | 2048 | 512 | 2048 | 2048 |
| Input embedding size ($C$) | 768 | 768 | 768 | 768 | 768 |
| Number of Process layers | 12 | 6 | 26 | 26 | 40 |
| Number of latents ($N$) | - | - | 256 | 256 | 256 |
| Latent size ($D$) | - | - | 1280 | 1280 | 1536 |
| FFW hidden dimension for latents | - | - | 1280 | 1280 | 1536 |
| Number of output queries during pretraining ($O$) | - | - | 512 | 2048 | 2048 |
| Dimension of learned queries ($E$) | - | - | 768 | 768 | 768 |
| FFW hidden dimension for outputs | - | - | 768 | 768 | 768 |
| Training steps/second | 7.3 | 2.9 | 7.4 | 7.6 | 4.2 |

Table 11: Perceiver IO architecture details for language experiments.

## F.3 MLM PRETRAINING

We pretrain all models on a mixture of the C4 dataset (Raffel et al., 2020) and English Wikipedia, where $70\%$ of the training tokens are sampled from the C4 dataset and the remaining $30\%$ from Wikipedia. We concatenate 10 documents before splitting into crops to reduce wasteful computation on padding tokens. We use the same masking strategy for SentencePiece and byte-level experiments: each word is masked independently with probability $15\%$ where word boundaries are defined using white-space boundaries.

The pretraining hyperparameters are given in Tab. 12. For the BERT (matching FLOPs) model trained on bytes, we reduce the model width from 768 to 512, the feed-forward hidden size from 3072 to 2048, the number of layers from 12 to 6 and the number of attention heads from 12 to 8. Given the longer sequence length of 2048 bytes, this model has about the same number of inference FLOPs as a BERT Base model on a sequence length of 512 tokens.

In order to decode, we use learned queries of the same dimension of the input array (Tab. 11). We have as many output queries as inputs to be able to predict the masked token at all positions in the sentence ($M=O$).

To get an insight into the learnt queries we visualize the attention weights in the first cross attention layer on a small paragraph (Fig. 7). We discover that the model has learnt both position and content based look-ups. The position-based look-ups can be either very sparse and precise or more distributed and periodic. This second mode appears somewhat less often and is more efficient because more data is being attended to at the same time, but also more distributed, since the values are subsequently averaged: this acts as a learned pooling. The content based retrievals focus mostly on syntactic elements like capital letters and punctuation (colon, exclamation marks, quotation marks, etc). This is probably because these are good word delimiters and can help the model reduce prediction uncertainty.

| | |
|---|---|
| Training steps | 500,000 |
| Batch size | 2048 |
| Masking strategy | Words |
| Optimizer | LAMB (You et al., 2021) |
| Learning rate | 0.00125 |
| Linear warmup steps | 1,000 |
| Cosine cycle decay | 500,000 |
| Weight decay | 0.01 |

Table 12: Hyperparameters for masked language modelling (MLM) pre-training experiments

## F.4 GLUE FINETUNING

Following Devlin et al. (2019), we specify a fixed-size hyperparameter grid and select the best dev performance across that grid for each task independently (Tab. 12). The full GLUE results are shown in Tab. 14. Following Devlin et al. (2019) we exclude the WNLI task. We use accuracy for all tasks expect STS-B and CoLA where we use Pearson correlation and Matthews correlation respectively. The average is computed by first averaging the results of MNLI-matched and MNLI-mismatched, which is then counted as a single task in the overall average.

For single-task experiments, we do not require a `[CLS]` token as we use a single decoding query vector. In both single-task and multi-task experiments an extra 2-layer MLP with a hidden size of $E$ and a tanh activation is used to map the the Perceiver IO outputs to the class logits (or regression target for STS-B).

## F.5 ABLATION ON THE NUMBER OF LATENTS

For a given FLOPs budget, there is a trade off between the number of latents $N$ and the width $D$ of the latents. We ablate this in Tab. 15 by varying the number of latents between 128, 256 (best), and 512. We adapt the latent dimension accordingly to match the FLOPs budget.

A bear walks into a restaurant. **He** tells his waiter, "I want a grilled…cheese." The waiter says, "What's with the pause?" "Whaddya mean?" the bear replies. "I'm a bear!"
A bear walks into a restaurant. He **tell**s his waiter, "I want a grilled…cheese." The waiter says, "What's with the pause?" "Whaddya mean?" the bear replies. "I'm a bear!"

(a) Very sharp location based attention.

A bear walks into a restaurant. He tells **his** waiter, "I want a grilled…cheese." The waiter says, "What's with the pause?" "Whaddya mean?" the bear replies. "I'm a bear!"
A bear walks into a restaurant. He tells his waiter, "I want a grilled…cheese." The waiter says, "What's with the pause?" "Whaddya mean?" the bear replies. "I'm a bear!"

(b) A more efficient and more distributed "periodic" location based attention.

A bear walks into a restaurant. He tells his waiter, "I want a grilled…cheese." The waiter says, "What's with the pause?" "Whaddya mean?" the bear replies. "I'm a bear!"
A bear walks into a restaurant. He tells his waiter, "I want a grilled…cheese." The waiter says, "What's with the pause?" "Whaddya mean?" the bear replies. "I'm a bear!"
A bear walks into a restaurant. He tells his waiter, "I want a grilled…cheese." The waiter says, "What's with the pause?" "Whaddya mean?" the bear replies. "I'm a bear!"

(c) Content based attention for syntactic elements like punctuation and capital letters.

Figure 7: Visualization of attention weights for a few queries in the initial cross-attention layer. We use the color to convey the weight of the attention and normalize by the maximum weight to make them easier to visualize. Best viewed in color.

| Training epochs | 10 |
| Batch size | {16, 32, 64} |
| Optimizer | LAMB |
| Learning rate | $\{1\times10^{-4}, 5\times10^{-5}, 2\times10^{-5}, 1\times10^{-5}\}$ |
| Linear warmup steps | 200 |
| Weight decay | 0.01 |

Table 13: Hyperparameters for GLUE finetuning experiments. We sweep over the values in brackets.

## G  POSITIONAL ENCODINGS FOR IMAGE AND AUDIO EXPERIMENTS

For all image experiments (with the exception of the ImageNet experiment that uses learned positions, Sec. A.1), we use a 2D Fourier feature positional encoding (Vaswani et al., 2017; Stanley, 2007; Mildenhall et al., 2020; Tancik et al., 2020) using a sine and cosine bands with frequencies spaced linearly from a minimum frequency to a maximum frequency. We use 64 sine/cosine bands per dimension in all settings. The minimum frequency is always set to the minimum frequency of the input signal, corresponding to a single full oscillation over the input dimension. The maximum frequency is typically set to the input's Nyquist frequency (e.g. 112 cycles for an image with 224 pixels per dimension). The input position used to construct the Fourier frequencies is scaled to [-1, 1] for each input dimension. For example, the upper left corner of an image is at position [-1, -1] while the bottom right corner is at position [1, 1]. We follow the same strategy using 1D and 3D Fourier feature positional encoding for audio's time and video's spatiotemporal inputs, respectively.

## H  OPTICAL FLOW: ADDITIONAL DETAILS AND RESULTS

Pre- and post-processing can provide non-trivial inductive biases when processing image data and also change computation time. In this section, we ablate these choices. The network in the main paper concatenates the two frames frames before extracting 3D patches around each pixel, each of size 3×3×2. Tab. 16 shows a few alternative designs for patch extraction. 1×1 means that only a single pixel (or pair of pixels) is used for each input element. 'Separate frames' means that the frames are not concatenated, but rather, input array elements are extracted independently from the

| Model | Tokenizer | Multi-task | CoLA | MNLI-m/mm | MRPC | QNLI | QQP | RTE | SST-2 | STS-B | Average |
|---|---|---|---|---|---|---|---|---|---|---|---|
| Bert Base (test) (Devlin et al., 2019) | SentencePiece | No | 52.10 | 84.60/83.40 | 84.80 | 90.50 | 89.20 | 66.40 | 93.50 | 87.10 | 80.95 |
| Bert Base (ours) | SentencePiece | No | 50.28 | 85.56/85.68 | 85.75 | 92.67 | 91.05 | 61.72 | 93.98 | 88.04 | 81.14 |
| Perceiver IO Base | SentencePiece | No | 47.11 | 84.53/85.03 | 87.25 | 92.12 | 90.22 | 65.23 | 94.38 | 88.18 | 81.16 |
| BERT (matching FLOPs) | UTF-8 Bytes | No | 20.06 | 74.11/75.55 | 77.00 | 85.75 | 88.23 | 53.91 | 89.00 | 82.84 | 71.45 |
| Perceiver IO | UTF-8 Bytes | No | 50.19 | 83.22/83.89 | 87.24 | 91.71 | 90.12 | 64.84 | 93.17 | 86.81 | 80.95 |
| Perceiver IO++ | UTF-8 Bytes | No | 52.54 | 84.13/84.91 | 86.03 | 92.06 | 90.46 | 66.54 | 93.98 | 87.93 | 81.76 |
| Perceiver IO (Shared input token) | UTF-8 Bytes | Yes | 47.43 | 82.03/82.65 | 89.58 | 90.18 | 89.20 | 82.03 | 93.17 | 77.95 | 81.49 |
| Perceiver IO (Task specific input token) | UTF-8 Bytes | Yes | 49.06 | 82.14/82.64 | 89.84 | 90.53 | 89.40 | 79.69 | 93.17 | 80.02 | 81.76 |
| Perceiver IO (Multitask query) | UTF-8 Bytes | Yes | 47.88 | 82.05/82.77 | 90.36 | 90.37 | 89.49 | 80.08 | 93.75 | 79.95 | 81.79 |

Table 14: Full GLUE results (higher is better). The first 3 models use SentencePiece tokens, the latter 3 use UTF-8 bytes directly.

| Number of latents ($N$) | Latent width ($D$) | FLOPs | Average GLUE score |
|---|---|---|---|
| 128 | 1920 | 120B | 75.84 |
| 256 | 1280 | 113B | 80.95 |
| 512 | 896 | 125B | 80.92 |

Table 15: Ablation on the UTF-8 Bytes Perceiver IO latent width versus depth.

two frames (thereby doubling the number of input elements). In the case of separate frames, $1 \times 1$ means essentially no preprocessing: each pixel becomes its own element with no spatio-temporal context whatsoever.

We also performed experiments with a less expensive input model which uses a $7 \times 7$ convolution to 64 channels, followed by a max pool, similar to the one used in our ImageNet experiments. After feeding this through the Perceiver IO architecture (including querying with the same convolutional features used as input), we have an output a feature grid with stride 4 and 64 channels, on top of which we apply a RAFT upsampling layer. This involves a linear projection from 64 dimensions to 2, which is the coarse-resolution optical flow estimate. We then upsample this flow for a given pixel in the high-resolution flow map by applying attention over a neighboring 3x3 block of the low-resolution flow map, following the uppsampling approach in RAFT (Teed & Deng, 2020).

We found that concatenating frames led to a non-trivial performance improvement across the more difficult Sintel.final and KITTI Flow 2015 (Menze & Geiger, 2015) datasets. Spatial context helps, and the impact of frame concatenation is larger when more context is available, suggesting that the algorithm is comparing spatial and temporal gradients. Convolutional downsampling and RAFT upsampling provide even more spatial context for both the input features and the queries, but this doesn't make up for the loss of resolution and overall performs slightly worse than using the full resolution.

Perceiver IO is somewhat slower on traditional GPUs than our baseline RAFT model, but we find that the trend reverses on TPUs, which is the target architecture for our work. For ease of comparison, we report inference speed on $1088 \times 436$ images, using a tiled inference setup. Our most expensive model achieves approximately 0.8 frames/sec on a 2017 TITAN Xp, and our lightweight model (with conv downsampling and RAFT-style upsampling) achieves 3.3 frames/sec, which is not far from the 10 frames per second reported for RAFT (Teed & Deng, 2020). On the publicly-available TPU v3, however, our most expensive model achieves 4.4 frames/sec on a single TPU core, and 17.8 frames/sec for the lightweight model. An efficient Tensorflow implementation of RAFT (Sun et al., 2020) (received courtesy of the authors) achieves only 1.6 frames/sec on the same hardware. We suspect that the difference is due to the gather operations required for RAFT but not for Perceivers, which are slow on TPU due to their poor memory locality properties.

Fig. 8 shows some results on example image pairs from the Sintel.final dataset. We see that the algorithm is capable of dealing with heavy occlusion, and can propagate optical flow across large regions with very little texture. The network can also deal with very large motions and very small objects.

Finally, to verify that Perceiver IO performs well on real-world data despite being trained only on synthetic imagery, we applied it to a small number (roughly 10) real videos taken from Getty images (www.gettyimages.com). Perceiver IO typically performs very well out-of-domain,

| Method | Patch size | Concat. frames | Downsample | Depth | Latents | Sintel.clean | Sintel.final | KITTI |
|---|---|---|---|---|---|---|---|---|
| PWCNet (Sun et al., 2018) | - | - | - | - | - | 2.17 | 2.91 | 5.76 |
| RAFT (Teed & Deng, 2020) | - | - | - | - | - | 1.95 | 2.57 | 4.23 |
| Perceiver IO | 3×3 | Yes | No | 24 | 2048 | 1.81 | 2.42 | 4.98 |
| Perceiver IO | 3×3 | No | No | 24 | 2048 | 1.78 | 2.70 | 6.19 |
| Perceiver IO | 1×1 | Yes | No | 24 | 2048 | 1.91 | 2.56 | 5.39 |
| Perceiver IO | 1×1 | No | No | 24 | 2048 | 1.72 | 2.63 | 5.93 |
| Perceiver IO | N/A | Yes | Yes | 24 | 2048 | 1.84 | 2.52 | 4.83 |
| Perceiver IO | N/A | No | Yes | 24 | 2048 | 1.90 | 2.53 | 6.66 |
| Perceiver IO | N/A | Yes | Yes | 16 | 1024 | 2.06 | 2.67 | 6.12 |

Table 16: Ablated Optical Flow results (end-point error, lower is better). The top Perceiver IO results show the configuration from the main paper. We ablate 1) patch size for the context surrounding each pixel, 2) whether the two frames are concatenated or input separately to the Perceiver, 3) whether the inputs and queries are downsampled by a factor of 4 using a convolution, and then subsequently upsampled with RAFT, and finally a the number of self-attention modules (depth) and number of elements in the latent array, resulting in a bottom-row network which is substantially less expensive than the original model.

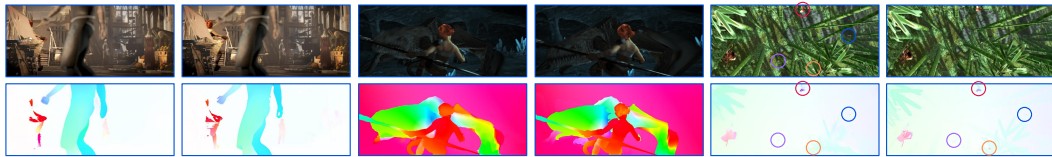

Figure 8: Qualitative examples of optical flow. For each image pair, we show the two frames (top), and then the estimated flow (bottom left) and the ground-truth flow (bottom right). In the left example, we see one person under heavy occlusion where the correct flow is propagated into a region with few details. Another person in the foreground has clothes with little texture and substantial blur, and yet the algorithm can propagate the flow across the entire region. In the center example, we see very large motions from both the dragon and the person, yet many fine structures are preserved like the pole. On the right, we see a forest scene with a few extremely small objects with very subtle motions (circled) which our algorithm is able to detect and segment correctly.

although some failure cases remain: for instance, shadows tend to be interpreted as objects (Autoflow contains no shadows), and large regions with compression artifacts but no other texture may result in hallucinated flow (Autoflow contains no video compression artifacts). We include three challenging examples in the supplementary zip file, each of which depict complex motion and small objects. Perceiver IO can pick up on remarkably small objects such as the water droplets thrown by the girl's shoe in `pigeon.mp4` or the confetti in `thai_dance.mp4`.

**Implementation details:** Our experiments with pixels and patches use a sine and cosine position encoding with 64 bands for both $X$ and $Y$, plus the raw $X$ and $Y$ values resulting in 258 extra features concatenated to the pixel or patch values. For experiments without concatenated frames, we have an additional time dimension which must be encoded with positional encoding, and for this we also use 64 sine and cosine bands (which are highly redundant, as there's only two frames). For this version, only the elements associated with the first frame are included as queries for the decoder. For both input and query, we project these concatenated features to 64 dimensions before inputting them into the transformer. We use a latent array with 2048 elements and 512 channels and 24 self-attention modules, each with 16 self-attention heads, unless otherwise noted. Our experiments with convolutional downsampling and RAFT upsampling use settings that are mostly similar, although we use no additional projection as the output of the convolutional network is already 64 channels. For these experiments, the output of the perceiver decoder's cross attend is 64 channels, which is fed into a RAFT-style upsampling operation. For the pixel- and patch-based models, total computational complexity for a forward pass on a $368 \times 496$ image is roughly 987 billion FLOPs, and there are roughly 27.9 million parameters.

In all cases, we train on the AutoFlow dataset (Sun et al., 2021), which consists of $400,000$ image pairs, for 480 epochs using a cosine learning rate schedule which starts at a learning rate of 4e-4. We use a batch size of 512. We use the LAMB (You et al., 2021) optimizer. We also use the default curriculum for AutoFlow, which gradually increases the severity of the augmentations over time. We find that naïve training on AutoFlow does not train, so we use an additional phase in this curriculum,

| Params | FLOPs (train) | FLOPs (eval) | Train steps/sec |
|--------|---------------|--------------|-----------------|
| 20.0M  | 310B          | 6.85T        | 4.4             |

Table 17: Additional details of the model used for Multimodal autoencoding.

where we completely disable all augmentations. Furthermore, for this phase, we feed every image pair twice in a batch: once forward, and once reversed. As the inverse flow is not currently available for AutoFlow, this inverse flow was computed via an approximation which averages all the flows terminating at a given pixel.

The evaluation datasets have a different resolution, so we evaluated in a tiled manner, using six evenly-spaced tiles. For pixels that are covered by multiple tiles, we average the predictions, weighted proportional the distance to the nearest edge of the respective tile (as we expect predictions nearer to the tile edges to be less accurate). We leave the possibility of making Perceiver IO invariant to input shape to future work.

# I    MULTIMODAL AUTOENCODING: ADDITIONAL DETAILS

For the multimodal autoencoding experiments, we patch preprocessing for both images and audio, and we embed the labels as one-hot labels. The patch size is $1 \times 4 \times 4$ for video and 16 for audio. The audio is sampled at 48kHz, or 1920 samples per frame. The decoder outputs $16 \times 224 \times 224 + 16 \times 1920/16 + 1$ vectors with 512 channels, that is, one element for each pixel in the video, one element for each audio patch, and one element for the classification label. These are then linearly projected to the appropriate channel size for each modality: 3 for videos, 16 for audio and 700 for classification (the logits for each of the 700 classes in Kinetics700). Finally, we un-patch the audio to arrive at the output audio. We note that we read and generate the audio waveform directly in the time domain; we do not transform first to a spectrogram.

We use a 387 dimensional 3D Fourier position embedding for each input video patch and a 385 dimensional 1D Fourier position embedding for each audio patch (385 to ensure the input dimensions to Perceiver IO match for all elements). In addition, we pad all input elements with a learned vector representing the modality; inputs from the same modality share the same token. In particular, we add a 317 dimensional modality embedding to video elements, a 319 dimensional modality embedding to audio elements, and a 4 dimensional modality embedding to the label, so that all elements have 704 features.

The decoder queries are also constructed from Fourier position embeddings for video and audio and a learned positional embedding for label: 387 features for video, 385 features for audio, and 1024 learned features for the label. We pad the queries for each modality with a different learned vector for each modality, so that the final feature size for the queries is 1026.

We train on Kinetics 700 (Smaira et al., 2020). We use batch size of 1024, and learning rate of 1e-3. The training loss is a weighted sum of the L1 loss for video, the L1 loss for audio, and the cross entropy loss for the label. The weightings are 0.03 for video, 1 for audio, and 0.0001 for the label; the loss weights are imbalanced in favor of audio because it is more difficult to obtain audio of high perceptual quality by directly outputting the waveform. We also tried a different weighting (0.03 for video, 1 for audio, and 1 for the label) to obtain higher classification accuracy. Additional model details are given in Tab. 17.

To help verify the quality of Perceiver IO's outputs on real-world data, we applied it a small number of real videos ($\sim$10) with audio taken from Getty Images. Perceiver IO is able to capture the structure of both video and audio inputs, despite encoding both jointly with a single network. The model introduces blurriness to both video and audio: this may be partially attributable to the preprocessing, which included coarse patching (Tab. 5) for both modalities due to the very high computational cost of processing raw video and audio inputs (which amount to over 2 million raw points). Although decoding can be done in parallel, allowing us to decode very large output arrays in sequential batches, Perceiver IO requires all points are encoded simultaneously. Addressing this limitation and scaling to even larger inputs is an important direction for future work.

