# OpenReview forum: "Perceiver IO: A General Architecture for Structured Inputs & Outputs"
_ICLR.cc/2022/Conference — ICLR 2022 Spotlight_

### Official Review · Reviewer_h233 · 2021-11-02

**Correctness:** 4
**Technical Novelty And Significance:** 3
**Empirical Novelty And Significance:** 3
**Recommendation:** 8
**Confidence:** 4

**Main Review:**

#### Strengths

- Major contribution is ability to scale the output to arbitrary length using an output query array. This enables supporting different types of output and scales well to different tasks.
- Byte level performance of Perceiver IO is impressive compared to BERT baseline. Byte level embeddings can be fed to Perceiver IO without any tokenization and this setting gets comparable performance with BERT sentence piece tokenization.
- Shows good results with multitask learning, using both single and task specific tokens.
- Authors do a thorough evaluation of this architecture on a wide variety of tasks, and achieves comparable results on multiple tasks in language, vision, multimodal, RL domains.

#### Weaknesses

- Novelty over Perceiver : Although I find this work quite impressive as it scales linearly with output embedding sizes as well as arbitrary types of outputs, the overall architecture seems incremental compared to Perceiver. One important change is there is no cross attention in the intermediate layers. I would love to see more detailed analysis why that is removed in this architecture and empirical evidence to support this change.

- How does choosing values for N,D affect the performance on various tasks. If we decouple input representation, output representation and base architecture, how does model performance scale with depth, N, D values.  Ablations on these hyper-parameters are required for more clarity.

- Given Perceiver can handle multiple modalities, I would be interested to see how it performs on multimodal tasks like VQA, Image/Text Retrieval, Hateful Memes etc and comparison to multimodal models like CLIP[1], ALIGN[2], SimVL[3].

  - [1] Radford, Alec, Jong Wook Kim, Chris Hallacy, Aditya Ramesh, Gabriel Goh, Sandhini Agarwal, Girish Sastry et al. "Learning transferable visual models from natural language supervision." arXiv preprint arXiv:2103.00020 (2021).
  - [2] Jia, Chao, Yinfei Yang, Ye Xia, Yi-Ting Chen, Zarana Parekh, Hieu Pham, Quoc V. Le, Yunhsuan Sung, Zhen Li, and Tom Duerig. "Scaling up visual and vision-language representation learning with noisy text supervision." arXiv preprint arXiv:2102.05918 (2021).
  - [3] Wang, Zirui, Jiahui Yu, Adams Wei Yu, Zihang Dai, Yulia Tsvetkov, and Yuan Cao. "SimVLM: Simple visual language model pretraining with weak supervision." arXiv preprint arXiv:2108.10904 (2021).


**Summary Of The Paper:**

This paper proposes a general purpose architecture(Perceiver IO) which can take any arbitrary type of inputs and produce arbitrary outputs. The architecture enables this while scaling linearly with input embedding size and output size. Perceiver IO performs comparably with SOTA results on a variety of tasks from language, vision and multimodal domains, which speaks about the generalizability of this architecture.

**Summary Of The Review:**

Perceiver IO extends the Perceiver architecture to scale to arbitrary outputs and output lengths. The proposed method is evaluated on a wide variety of tasks and achieves good results comparable to SOTA in most of them. I recommend the current score, with minor reservations over added novelty over Perceiver and also I would encourage the authors to include more analysis and comparison with other SOTA multimodal models.

---

> ### Author Response · Authors · 2021-11-23
> **Response to Reviewer h233**
>
> We thank the reviewer for the helpful review. We address open questions and remarks below:
>
> > One important change is there is no cross attention in the intermediate layers. I would love to see more detailed analysis why that is removed in this architecture and empirical evidence to support this change.
>
> We did not use iterative encoder cross-attention because the input cross-attention was the single most expensive component of the original Perceiver architecture. Each encoder cross-attention is expensive because it has complexity linear in the input size (see section D.2), which is very large for some domains (see Tables 6 and 7). Because of this, iterative cross-attention is disproportionately expensive and led to marginal improvements in early experiments on both language and ImageNet settings. Only a single encoder cross-attention is necessary to use the efficient encode-process-decoder scheme described in the paper.
>
> Quantitatively, the best-performing ImageNet architecture described in the Perceiver paper reached 78.0 top-1 while requiring 707B FLOPs, while the Perceiver IO model trained in the same input configuration (2D Fourier Features) reaches 79.0 top-1 accuracy while requiring around half as many FLOPs (404B). In early experiments, we confirmed that a similar pattern held on masked language modeling experiments: when matched for FLOPs, models that were deeper but used only a single cross-attend outperformed models that were shallower but used iterative cross-attentions. Because iterative cross-attention gave little to no performance difference while requiring a large increase in FLOPs, we focused on models with the simpler, compute-efficient model described in the paper on all the rest of the 8 domains described here.
>
> > Given Perceiver can handle multiple modalities, I would be interested to see how it performs on multimodal tasks like VQA, Image/Text Retrieval, Hateful Memes etc and comparison to multimodal models like CLIP[1], ALIGN[2], SimVL[3].
>
> Thanks a lot for this great suggestion. We believe PerceiverIO is indeed quite tailored to multimodal applications as shown by our experiments on video and audio in the paper. The suggested applications make a lot of sense and we would be thrilled to see PerceiverIO excel in those tasks. But it would require a non-trivial amount of work to set up experiments on any of these domains and hence believe these are beyond the scope of this paper but would definitely constitute a great avenue for future work.
>
> > how does model performance scale with depth, N, D values
>
> We generally found that models performed better with increased depth, number of latents (N), and latent channel size (D) performed better but also were more expensive and more susceptible to overfitting. On masked language modeling, a FLOPs-controlled experiment on the number of latents (N) and the channel width (table 14) showed that the sweet spot for N was around 256, with slight drops in performance for a smaller number of latents. On ImageNet, the results of our experiments mirrored those in the Perceiver paper (Jaegle et al 2019, figure 5): model performance improves for all three, but tends to saturate or lead to greater overfitting beyond the configurations reported in the paper.

---

### Official Review · Reviewer_MYnu · 2021-11-02

**Correctness:** 3
**Technical Novelty And Significance:** 3
**Empirical Novelty And Significance:** 4
**Recommendation:** 8
**Confidence:** 3

**Main Review:**

The main strength of Perceiver IO is its high flexibility to various input domains. While keeping the main architecture, we can focus on engineering the input query preprocessing with domain-specific knowledge. Perceiver IO not only shows flexibility but also shows high performance. The results are comparable to or sometimes outperform previous domain-specific methods. For example

- Table 1 shows that Perceiver IO works well even without tokenization, while BERT + UTF-8 bytes show a significant performance drop (81.0 -> 71.5). It may show that Perceiver IO can perform well even without minimal domain knowledge (tokenization).
- Table 2 shows that the input query engineering can improve the target task (e.g., single-task query to 8 multi-task queries).
- Table 5 shows that the small input preprocessing engineering can improve the performance a lot. For example, Perceiver IO shows 82.1 top-1 accuracies for conv preprocessing while Perceiver IO with Fourier features shows 79.0.

I have minor concerns about ImageNet experiments.

- Although this paper is not aiming to state-of-the-art ImageNet performances, I think Perceiver IO is not yet comparable to state-of-the-art models on ImageNet image classification (as described in the last paragraph of Section 1). Perceiver IO still needs tremendous FLOPs compared to vision-specific models (Table 17). For example, ResNet shows 78.6 top-1 accuracy with 4.1 GFLOPs, while Perceiver IO Fourier shows 79.0 top-1 accuracy with 407 GFLOPs (99 times larger FLOPs). Stronger Perceiver IO (conv) shows 82.1 top-1 accuracy with 369 GFLOPs, where CaiT-M48 448 shows 86.5 top-1 accuracy and 329.6 GFLOPs.
- "Perceiver IO (pretrained)" and "Perceiver IO" in 2D Fourier features are different models. For example, "Perceiver IO (pretrained)" does not use weight sharing, and uses larger channel sizes. As a result, Perceiver IO (pretrained) has 4.38 times larger parameters compared to Perceiver IO (but half GFLOPs). The current text can make a reader mislead that "Perceiver IO (pretrained)" is the same model as "Perceiver IO", but the only difference is the JFT pretraining, which is not true.

> Unlike in the other ImageNet experiments, we do not share weights in the latent self-attention process modules, but use a 16-layer latent network with no weight sharing in depth. Unlike the other ImageNet experiments, the process-module MLPs use a hidden layer with 4× the number of channels (rather than 1× as on other ImageNet experiments)

To avoid confusion, I suggest replacing Table 5 with Table 17. Also, I would like to suggest to clarify that "Perceiver IO (pretrained)" and "Perceiver IO" are not the same model.

## Questions

- Perceiver IO shows better performances consistently compared to Perceiver IO baseline on GLUE (Table 13), Audio Set (Table 7), and ImageNet (Table 17). To my knowledge, there is no specific reason to the proposed output array cross-attention works better than the conventional classification head. I wonder what is the opinion by the authors what is the source of the improvements.
- In Table 5, I found that using 2D + maxpool preprocessing harms Perceiver performances (78.6 -> 77.4), but it helps Perceiver IO a lot (79.0 -> 82.1). As far as the reviewer understood, the main difference between Perceiver and Perceiver IO is the last decoding layer, not the input processing. I wonder what is the opinion of the authors why Perceiver and Perceiver IO show contradictory results on ImageNet Fourier features and conv preprocessing.
- I wonder why there is no Perceiver IO (conv) + pretraining result. Do the authors have any plan to add Perceiver IO (conv + pretrained)? (this is not a mandatory experiment for the rebuttal)

## Minor comments

- Many details are missing in the main text. To understand the method, the readers should read through the heavy appendix and the previous Perceiver paper. I fully understand that this is because of the page limitation, but this paper is somewhat hard to understand at first glance without any prior knowledge.
- In my opinion, it would be very helpful to readers if Table 8 and 9 are in the main text.
- pg 3. Typo -- unnecessary space *( "inducing points")*
- pg 29. Missing hyperlink (Tab. ??)
- In my opinion, it will be very interesting if there is one unified Perceiver model pretrained on various domain data. For example, one can train a Perceiver model trained by language and vision modalities, where each modality has different input and output arrays.

**Summary Of The Paper:**

This paper proposes a general-purpose neural network architecture named Perceiver IO. With only small modifications on the query side, Perceiver IO can handle various inputs, such as languages, images, videos, point clouds, optical flows, or game agents. Perceiver IO is hugely dependent on the previous work, Perceiver (Jaegle et al.). The main difference between Perceiver and Perceiver IO is the final decoding module. Instead of using a classification head, Perceiver IO uses an additional cross-attention module with an output array. For example, if a user wants to make Perceiver IO be a classification model, the output array becomes 1 X # classes. Perceiver IO shows impressive results in the experiments compared to domain-specific designed architectures in various domains (language, optical flow, audio-video autoencoding, image classification).

Perceiver IO shows its flexibility on various input domains and domain knowledge. For example, Perceiver IO shows even comparable performances to the BERT baseline on the GLUE benchmark without the conventional tokenization but with only UTF-8 bytes (Perceiver IO with tokenization shows 81.1 and Perceiver IO with UTF-8 bytes shows 81.0 while BERT baseline with comparable FLOPs shows 81.1). On the other hand, in the vision domain, by using 2D convolution and max-pooling preprocessing, Perceiver IO shows better top-1 accuracy compared to 2D Fourier features (82.1 for conv preprocessing and 79.0 for 2D Fourier features).


**Summary Of The Review:**

This paper provides very a strong empirical contribution to a unified architecture for various input domains. Not only showing flexibility, but Perceiver IO also shows powerful performances in various domains. Although I have a small concerns that ImageNet results (a reader can mislead the results) I think this paper is a good paper and indeed recommend acceptance.

--------------------
Post-rebuttal comments.

The authors addressed my concerns very well in the rebuttal. I will keep my decision as "8: accept, good paper".

---

> ### Author Response · Authors · 2021-11-23
> **Response to Reviewer MYnu**
>
> We thank the reviewer for the helpful review. We have incorporated all typographical and formatting suggestions to the extent possible with the 9 page limit. We managed to fit Table 17 into the main paper. We could unfortunately not fit Tables 8 and 9 into the main paper, but to raise visibility of these tables, we moved them forward (they are now Tables 6 and 7) and we now point the reader to them right away in Figure 1.
>
> We address open questions and remarks below:
>
> > Although this paper is not aiming to state-of-the-art ImageNet performances, I think Perceiver IO is not yet comparable to state-of-the-art models on ImageNet image classification (as described in the last paragraph of Section 1).
>
> We have rephrased the discussion of the ImageNet results to better contextualize our claims and make it completely clear that we are not claiming SoTA performance here. As the reviewer suggests, these experiments are intended to illustrate the flexibility in how the architecture can be applied while still producing good results.
>
> > The current text can make a reader mislead that "Perceiver IO (pretrained)" is the same model as "Perceiver IO", but the only difference is the JFT pretraining, which is not true.
>
> We thank the reviewer for pointing this out. The revised manuscript now refers to the two architecture configurations as Perceiver IO, configuration A and Perceiver IO, configuration B (pretrained) and we have clarified the difference in the caption. We have replaced table 5 with table 17 as suggested.
>
> > To my knowledge, there is no specific reason to the proposed output array cross-attention works better than the conventional classification head. I wonder what is the opinion by the authors what is the source of the improvements.
>
> We thank the reviewer for this suggestion. We believe this is because the attentional decoder is more expressive than a standard average + project decoder. This expressivity gain comes for two reasons. First, the attentional decoder learns to weight the contribution of each latent to the output in a data-dependent manner before averaging them, unlike the average + project decoder, which weights the contribution of all latent equally. Second, the attentional decoder includes an MLP after the pooling step (as is standard for attention modules), while typical average + project decoders use a linear layer after pooling. We believe these features allow the decoder to learn a better pooling strategy in practice. We have included a new figure to illustrate the difference between the two decoders and to give a better intuition for why a cross-attention decoder might work better (Appendix Figure 6 in the updated manuscript).
>
>
> > In my opinion, it will be very interesting if there is one unified Perceiver model pretrained on various domain data
>
> We agree that this is a very promising direction for future work opened up by the architecture developed here, but it involves solving other technical challenges and is beyond the scope of the current paper (see reply to Reviewer 2doh).
>
> > Do the authors have any plan to add Perceiver IO (conv + pretrained)? (this is not a mandatory experiment for the rebuttal)
>
> We are currently running this experiment, but it has not converged due to difficulty scheduling the required resources. We will describe the results when they are available.

---

> > ### Comment · Reviewer_MYnu · 2021-11-29
> > **Thanks for your responses**
> >
> > Dear authors,
> >
> > The authors addressed my concerns very well in the response. I will keep my decision as "8: accept, good paper". Thanks for the great work!

---

> > ### Author Response · Authors · 2021-11-29
> > **Results for Perceiver IO (conv + pretrained)**
> >
> > The additional pretraining+fine-tuning experiment has finished running, and it produces **86.4%** top-1 eval accuracy on ImageNet. This model is identical to the configuration B model used for 2D Fourier Feature (FF) pretraining other than the initial layers (i.e. the addition of the initial conv + max-pool). We found that we had to lower the learning rate during fine-tuning to produce good results, from 2e-3 for 2D FF to 2e-4 for conv+maxpool.
> >
> > When pretraining, adding the conv+maxpool layers leads to a 1.9% top-1 boost over using 2D Fourier features alone (86.4 vs. 84.5), compared to the 3.1% boost when training only on ImageNet (82.1 vs. 79.0). Using conv+max leads to a modest reduction in FLOPs and essentially the same number of parameters: the pretrained model with conv+max uses 176B FLOPs (vs. 213B for 2D FF) and 212.38M params (vs. 212.37M  for 2D FF).
> >
> > We will update the final manuscript to incorporate these results.

---

### Official Review · Reviewer_qg6s · 2021-11-03

**Correctness:** 4
**Technical Novelty And Significance:** 3
**Empirical Novelty And Significance:** 2
**Recommendation:** 8
**Confidence:** 3

**Main Review:**

Pros:

1. This paper introduced a generic architecture for coping with different tasks with various input and output lengths. It is different from Transformer architecture and Perceiver architecture in that it uses a latent process to compress the larger number of input tokens into a hidden space, and then decode the output from this hidden space. This way it can decouple the inputs and outputs which has the potential to significantly reduce the cost brought by a large number of input and output tokens.

2. The authors presented a thorough study of Perceiver IO across different application scenarios, such as language, vision, multi-modality. Extensive experiments demonstrate that the proposed method can achieve comparable or even better performance than the baselines. To facilitate the adaptation, the authors suggested a number of good ways to convert the inputs/outputs into certain formats.

Cons:

1. The authors should list the sizes of inputs, latent, and output arrays for each of the downstream tasks to give the audience a better sense of the complexity. For some of the tasks, such as ImageNet classification, the authors should also report the FLOPs in comparisons with prior arts, e.g., ViTs.

2. The main merit of Perceiver IO compared with Perceiver is that it further introduces a cross-attention-based output decoder on top of the latent arrays. Though Perceiver was originally proposed for classification tasks, it can be also used to cope with structured outputs. For some of the tasks except for ImageNet classification listed in this paper, I am curious whether Perceiver can be applied and how it performs compared with Perceiver IO.

3. Perceiver IO is proposed as a generic architecture for various tasks by modeling them as a read-process-write process. One question is that whether we should reformulate all these tasks as the way shown in this paper. This paper does show some encouraging results on various tasks compared with the baselines. However, they are only compared with baseline methods and still underperforms established methods in specific domains. As we know, different tasks can still benefit a lot from the specific domain knowledge (e.g., 2D spatial for image recognition). Actually, we can also think of the Transformer encoder as a generic architecture in numerous domains. Then, do we need to build up a higher-level generic architecture upon the Transformer encoder?

4. A good part of unifying architectures for various tasks is that we can train the same set of parameters using different tasks. In this paper, the authors demonstrated the effectiveness of Perceiver IO across different settings, I am wondering whether such a generic architecture can be used for multi-task training so that it can leverage the training data from different tasks.

**Summary Of The Paper:**

In this paper, the authors proposed a new general architecture called Perceiver IO for various tasks with different types of inputs and outputs. The Perceiver IO employes a read-process-write architecture, in which input arrays are first projected to the latent space through cross-attention and then the outputs are generated by querying the latent space through some output query arrays. Such a generic pipeline can be applied to various tasks spanning from single modality language tasks, multi-modal tasks, dense prediction tasks, and even symbolic predictions. The main benefit of the proposed Perceiver IO is that it decouples the inputs and outputs using a latent processing module so that it can cope with the arbitrary length of input arrays while outputting an arbitrary number of predictions. Extensive epxeriments are conducted on language understanding and masked language learning, optical flow estimation, image classification, multi-modal autoencoding, etc, and showed that the proposed Perceiver IO can achieve comparable performance to strong baselines in different domains.

**Summary Of The Review:**

Overall, I like the idea of Perceiver IO considering it is a good way of modeling arbitrary numbers of input and output tokens. Based on the experimental results on various downstream tasks, it indeed shows some superiorities over the strong baselines. The experiments are solid enough to justify the main claim. However, as discussed above, my main question is that whether we want to develop a unified read-process-write architecture for all these tasks. What are the main merits of this act? I will hold my score to the borderline and wait for more feedback from the authors.

[Post-rebuttal comments]

After reading the authors' feedbacks to all reviewers including myself, I think they had addressed most of the concerns and questions. All reviewers think this work is novel and appreciate its solid contributions. As such, I raised my rating to "accept", and look forward to the furture work of applying it to learn from multiple tasks.

---

> ### Author Response · Authors · 2021-11-23
> **Response to Reviewer qg6s**
>
> We thank the reviewer for the helpful review. We address open questions and remarks below:
>
> > do we need to build up a higher-level generic architecture upon the Transformer encoder? … What are the main merits of this act? … should reformulate all these tasks as the way shown in this paper[?] …
>
> Transformers are general purpose in principle but don’t scale well enough with input/output size to be practical on many domains. Transformer of any practical depth are impractical to scale beyond input and output sequence lengths in the thousands. This makes them hard to apply to problems like optical flow estimation, which has hundreds of thousands of inputs/outputs. Transformers’ poor scaling means that preprocessing is needed to reduce the input sequence length even on language (via tokenization), where Transformers were initially applied. The encoder-process-decoder architecture used by Perceiver IO preserves the general purpose characteristics of Transformers, but scales much larger inputs and outputs (see section 3.1 for a discussion and see Table 6 for the size of domains considered here, which range from the hundreds into the hundreds of thousands).
>
> A general purpose architecture is desirable because it makes it easier to work on domains with many kinds of inputs (e.g. video+audio+label autoencoding, AudioSet) and other domains where we don’t have good inductive biases or domain-specific architectural priors. The existence of a domain general architecture makes it easier to get started on problems where standard architectures don’t apply, because it mitigates the need to develop new architectures from scratch to start on a domain. Perceiver IO in particular also allows domain-specific inductive biases to be incorporated if they’re available - for example using mel-spectrogram preprocessing for AudioSet, using 2D convolutions and max pooling or 2D-aware position encodings for ImageNet. We explore these kinds of configurations extensively in the paper: see Table 6 for an overview.
>
> This flexibility means that even if domain information (like 2D structure) is desirable for a given application, a general purpose architecture can reduce the implementation and maintenance overhead of a research codebase, because the main architecture itself can be reused and optionally coupled with domain-specific auxiliary components. In practice, Perceiver IO’s scaling properties allow us to simplify standard input pipelines - removing tokenizers in language, simplifying architectural pipelines used in optical flow, even allowing us to remove 2D information from image classification.
>
> As the reviewer points out, a general purpose architecture like Perceiver IO also opens up new research directions, including training a single set of weights on domains like those we address here individually. Solving this problem involves other problems, including multidomain loss balancing and engineering challenges around handling data for problems with data of very different shapes and sizes. We agree this is an important area for future research but is beyond the scope of the current paper, which focuses on developing and evaluating the Perceiver IO architecture itself.
>
> > The authors should list the sizes of inputs, latent, and output arrays for each of the downstream tasks to give the audience a better sense of the complexity
>
> We provided this information in the original submission, but unfortunately were unable to fit it in the main paper: see tables 6 and 7 of the appendix for all of these numbers (these were Table 8 and 9 in the original submission, see response to Reviewer MYnu).
>
> > should also report the FLOPs in comparisons with prior arts, e.g., ViTs.
>
> See table 5 for a comparison of FLOPs on ImageNet for Perceiver IO and representative models from the literature. These numbers were previously included in Table 17 in the appendix, but we have moved them to the main paper as suggested by Review MYnu. We also compare FLOPs to baselines on masked language modeling (table 1) and StarCraft II (table 6), and we discuss model complexity and speed for optical flow vs. the previous state-of-the-art methods in section G.
>
> > For some of the tasks except for ImageNet classification listed in this paper, I am curious whether Perceiver can be applied and how it performs compared with Perceiver IO.
>
> Perceiver uses a decoder that makes it suitable for only classification and regression tasks. Of the tasks considered here, it can be applied to ImageNet and AudioSet classification, and we include comparisons between Perceiver and Perceiver IO in both of these domains (see Table 7 for AudioSet results).
>
> We discuss this point briefly in the introduction, and to make the point more clearly we have added an additional diagram illustrating the difference between the Perceiver and Perceiver IO decoder, with a discussion of the limitations of the original average + project decoder (Figure 6 of the appendix. See response to Reviewer MYnu).

---

> > ### Comment · Reviewer_qg6s · 2021-11-29
> > **Thanks for the feedbacks**
> >
> > I would like to appreciate the comments from the authors on the raised questions.
> >
> > Particularly, the authors did a good job addressing the main concern that whether we need a unified architecture like Perceiver IO for different tasks in different modalities.  I totally agree with the authors that Perceiver IO is very good at coping with long-sequence inputs and/or outputs that were not tractable with conventional Transformer. As such, it is superior to Transformer when being applied to language modeling, optical flow estimation, autoencoding, etc. Nevertheless, I am still conservative to position it as a better choice for classification tasks like image classification, video+audio classification, knowing that many multi-scale architectures (e.g., Swin) showed strong results. Considering this work is mainly focusing on a novel and generic architecture and demonstrating its effectiveness (not SoTA) in various scenarios, I think it has already done a great job. In general, I am still interested in what if applying Perceiver IO on a union of multiple tasks of different modalities, as also pointed out by other reviewers. I believe this will be a very exciting future work.
> >
> > I also appreciate the authors for their efforts to address the concerns raised by other reviewers. After reading through the other reviewers' comments and the authors' feedbacks, I can sense all reviewers appreciate the novelty of this work, the thorough experimental results, and the solid technical contributions. As a result, I am very happy to raise my rating to "accept", hoping it can inspire many future works.
> >
> > Thanks again for the great efforts.
> >
> > thanks,

---

### Official Review · Reviewer_2doh · 2021-11-04

**Correctness:** 4
**Technical Novelty And Significance:** 4
**Empirical Novelty And Significance:** 4
**Recommendation:** 8
**Confidence:** 3

**Main Review:**

#### Strength
- The idea of perceiver IO is novel and solid -- a general architecture capable of handling general-purpose inputs and outputs across different tasks and modalities. This is very promising to simplify the construction of highly tuned task-specific neural pipelines and improve the multimodal and multi-task problems.

- Each component in perceiver IO is necessary and well defined for the proposed tasks. 2D byte array input enables the task agnostic input for different modalities. Perceiver network that scales linearly with the size of input and output, and non-auto regressive decoding make the decoding to raw output possible.

- The proposed architecture is tested on massive experiments including language understanding tasks, optical flow, video audio class autoencoding, image classification, and starcraft II and achieves superior performance. Each task is supported with a detailed ablation study to shed light on future research.

#### Weakness
- In Table 1, the WNLI task is excluded from the GLUE benchmark, I wonder what is the reason this task is removed?

- The non-autoregressive decoding enables fast decoding for large outputs. However, for tasks that require more information on prior decoding context (machine translation or image generation tasks), does the proposed model can still perform well on those tasks?

- Although perceiver IO removes the task-specific pre-processing (tokenization, patch embedding, etc.), the model still requires huge engineering efforts to adapt the model for different tasks. For example, the model hyper-parameters are quite different for different tasks.

- One huge benefit of perceiver IO is to train different tasks together and explore the transfer between different tasks/modalities, which is not explored in this paper.

**Summary Of The Paper:**

This paper proposed perceiver IO, which is a general architecture for structured input & output. Perceiver IO is based on the perceiver architecture which scales linearly with the size of inputs and outputs and augments with a flexible querying mechanism similar to NeRF. Different from prior work, perceiver IO directly operates in the raw input space -- UTF-8 bytes for language, xy coordinates in optical flow, raw audio, etc. The output can be arbitrary in size and different structures, and with non-autoregressive decoding, the model can handle large input and output sizes. The proposed model is tested on variety of tasks, including language modeling, optical flow, video audio class autoencoding, image classification, and starcraft II, and achieves superior performance.

**Summary Of The Review:**

The paper is well written, the idea is solid and novel, supported with massive and strong experimental results. Overall, the is a strong submission and I would recommend accepting the paper.

---

> ### Author Response · Authors · 2021-11-23
> **Response to Reviewer 2doh**
>
> We thank the reviewer for the helpful review. We address open questions and remarks below:
>
> >In Table 1, the WNLI task is excluded from the GLUE benchmark, I wonder what is the reason this task is removed?
>
> We follow the protocol of the original BERT paper in excluding the WNLI task (as noted in Table 1 and footnote 8 of https://arxiv.org/abs/1810.04805). The GLUE Benchmark website (https://gluebenchmark.com/faq) contains additional information on why WNLI is excluded (FAQ question 12): (1) the train and dev sets contain identical sentences with opposite labels and (2) the test set is drawn from a different distribution than train and dev sets. Both of these features make evaluation on this dataset problematic.
>
> > The non-autoregressive decoding enables fast decoding for large outputs. However, for tasks that require more information on prior decoding context (machine translation or image generation tasks), does the proposed model can still perform well on those tasks?
>
> We thank the reviewer for this very good question. For generative tasks such as machine translation, image generation or language modeling auto-regressive models are indeed the current dominant paradigm.
>
> We could apply Perceiver IO in an auto-regressive manner at inference time to sample each token in turn, but there is no straight-forward way to efficiently train the Perceiver IO in that regime. Indeed standard Transformers can be trained auto-regressively by using causal masks in the self-attention so that past tokens cannot attend to future ones. Coupled with teacher-forcing, this technique allows Transformers to be trained on all input tokens with a single forward-backward pass, which makes the training efficient. Perceiver IO however introduces a latent bottleneck in order to scale to long sequences. The Perceiver IO latents have access to the full input and because there are fewer latents than inputs, it is not straight-forward to impose a causal structure on them.
>
> We believe this is an important direction for further research that may enable scaling generative models to very long sequence lengths (e.g. for video generation or book-length language modeling), but this is well beyond the scope of our current work.
>
> >Although perceiver IO removes the task-specific pre-processing (tokenization, patch embedding, etc.), the model still requires huge engineering efforts to adapt the model for different tasks. For example, the model hyper-parameters are quite different for different tasks.
>
> It’s true that Perceiver IO’s hyperparameters need to be adapted to each domain to produce the best possible performance. However, compared to alternative approaches to solving the full set of tasks we address here - which span optical flow, language understanding, multimodal processing, etc. - tuning Perceiver IO requires less overall engineering effort than baseline approaches. This is because each of these domains would typically be handled by a different, specialist architecture. Because we use a single base architecture for each domain, we can tune the architecture to a new domain by sweeping essentially the same set of hyperparameters used on other domains, rather than selecting architectures per domain and tuning different hyperparameters for an entirely different architecture.
>
> >One huge benefit of perceiver IO is to train different tasks together and explore the transfer between different tasks/modalities, which is not explored in this paper.
>
> We agree that training in a multitask setting is one of the main benefits of Perceiver IO. We in fact do this in the multitask language experiments (the last paragraph of Section 4.1). However, we do not exhaust the full potential of transfer capabilities of Perceiver IO. In particular, it would be interesting to compare this with the transfer capabilities of various different architectures and to explore multidomain multitask training. But each of these problems requires additional technical problems be solved and we leave this question for further work.

---

### Author Response · Authors · 2021-11-23
**Changes to original manuscript**

- Added Figure 6 and Section D.3 describing the relationship between attention decoder average + project decoder
- Added reference to Fig 6 and Sec. D.3 in sections 4.4 and B.
- Moved Tables 7 and 8 (summary tables) ahead of the StarCraft Table. New labels: Table 7 -> Table 6, Table 8 -> Table 7, Table 6 -> Table 8.
- Added reference to Tables 6 and 7 in Figure 1.
- The original Table 17 replaces the original Table 5.
- Fixed broken links in appendix.
- Reworded ImageNet section to make claims clearer.
- Standardized table formats.
- Added missing lines for AudioSet to table 9.
- Fixed AudioSet input channel dimensions in table 8.
- Fixed ImageNet learned position input channels in Table 8.
- Corrected flow feature sizes in Table 9.
- Added AudioSet FLOPs to table 7 and updated the caption.
- Section 1: removed a comma: “to a fixed-size latent space, that is further processed” -> “to a fixed-size latent space that is further processed”

---

### Decision · Program_Chairs · 2022-01-20

**Decision:**

Accept (Spotlight)

**Comment:**

This paper proposes Perceiver IO, a general neural architecture that handles general purpose inputs and outputs. It operates directly in the raw input domains, and thus does away with modality specific architecture components. The paper contains extensive experiments showing the capabilities of this architecture in different domains. The paper received very positive reviews from all reviewers. Some concerns included a need for additional details such as a missing task from GLUE, FLOPs comparisons to past works, nomenclature for the versions of Perceiver IO, etc. These concerns were well addressed by the authors. Others concerns by reviewers were the lack of experiments in a multi task setting. However, it was acknowledged by the authors and reviewers that this is an open area of research and is a good fit for future work. Given this high quality submission, strong reviews and a very positive discussion amongst authors and reviewers, I recommend accepting this paper.